# LIMEFLDL: A Local Interpretable Model-Agnostic Explanations Approach for Label Distribution Learning

Xiuyi Jia [1]    Jinchi Li [1]    Yunan Lu [1 2]    Weiwei Li [3]

## Abstract

Label distribution learning (LDL) is a novel machine learning paradigm that can handle label ambiguity. This paper focuses on the interpretability issue of label distribution learning. Existing local interpretability models are mainly designed for single-label learning problems and are difficult to directly interpret label distribution learning models. In response to this situation, we propose an improved local interpretable model-agnostic explanations algorithm that can effectively interpret any black-box model in label distribution learning. To address the label dependency problem, we introduce the feature attribution distribution matrix and derive the solution formula for explanations under the label distribution form. Meanwhile, to enhance the transparency and trustworthiness of the explanation algorithm, we provide an analytical solution and derive the boundary conditions for explanation convergence and stability. In addition, we design a feature selection scoring function and a fidelity metric for the explanation task of label distribution learning. A series of numerical experiments and human experiments were conducted to validate the performance of the proposed algorithm in practical applications. The experimental results demonstrate that the proposed algorithm achieves high fidelity, consistency, and trustworthiness in explaining LDL models.

## 1. Introduction

Learning with ambiguity has emerged as an important research direction in the fields of machine learning and data mining in recent years. There are currently several paradigms for dealing with label ambiguity, such as multi-label learning (Tsoumakas & Katakis, 2007), multi-instance learning (Carbonneau et al., 2018), and label distribution learning (LDL) (Geng, 2016). Among these, LDL stands out as an innovative learning paradigm that not only answers which labels are relevant to the sample but also further characterizes the relative importance of labels to the sample. LDL handles multi-label samples where labels exhibit co-occurrence patterns (simultaneous changes) and dependency propagation (one label's presence affects others' distributions). Given that LDL can provide richer semantic information, it has been widely applied in practical application scenarios such as disease diagnosis (Wang et al., 2022), emotion recognition (Jia et al., 2019), etc. Existing LDL research primarily focuses on the design and optimization of algorithms, but the interpretability of LDL algorithms have been neglected.

Interpretability plays a crucial role in understanding the decision-making logic of models on given data, helping researchers optimize models more effectively, and promoting the application of models in real-world scenarios. Within the existing field of interpretable machine learning, the LIME (local interpretable model-agnostic explanations) algorithm (Ribeiro et al., 2016) is widely used to enhance the interpretability of models. This is achieved by generating simple models to approximate the local behavior of black-box models. However, the LIME algorithm is designed for single-label learning tasks and focuses on the local region of a single label, making it unsuitable for direct application to LDL tasks that involve multiple interdependent real-valued labels. One intuitive approach is to apply the LIME algorithm in parallel to LDL tasks, but this approach faces significant challenges: (1) In LDL tasks, there are complex dependencies and interactions among labels. In such cases, the single-label explanations provided by LIME can be misleading due to the disregard for inter-label correlations. For example, in medical diagnostic tasks, "pneumonia" and "acute respiratory distress syndrome" often need to be addressed simultaneously; a parallel explanation might overestimate the risk of pneumonia alone and underestimate the risk when both pneumonia and acute respiratory distress syndrome are present. (2) The approach encounters compu-

[1]School of Computer Science and Engineering, Nanjing University of Science and Technology, China [2]Department of Computing, The Hong Kong Polytechnic University, China [3]College of School of Computer Science and Technology, Nanjing University of Aeronautics and Astronautics, China. Correspondence to: Weiwei Li <liweiwei@nuaa.edu.cn>.

*Proceedings of the 42nd International Conference on Machine Learning*, Vancouver, Canada. PMLR 267, 2025. Copyright 2025 by the author(s).

tational complexity issues; when the dataset is very large, generating a large number of perturbed samples for a single label and training local models becomes extremely time-consuming. (3) The local solution of parallelized LIME tends to violate the constraints of label distribution, which can lead to invalid label description degrees and unreliable explanations. For instance, when the true description degree of a label is very low, it can be predicted as a negative value, an invalid label description degree.

To address these challenges, we propose an improved LIME method for the local interpretability issues in LDL, named LIMEFLDL. Within this interpretive framework, (1) to explain the dependencies among labels, we design a feature attribution matrix that forms a feature attribution distribution for each feature. This distribution considers the impact of features on the entire label distribution and can be used to explain the inter-label dependencies, resulting in more consistent explanations. (2) To reduce computational complexity, we minimize the distance between the black-box model and the explanation model's output within the generated local region in the form of a label distribution. This process requires only one local sampling and does not necessitate generating a local region for each label, significantly reducing computational complexity. Additionally, we also provide a feature selection scoring function suitable for the label distribution format to accelerate the efficiency of explanation generation. (3) To avoid violating label distribution constraints, we incorporate linear constraints and penalty function constraints within the local model, ensuring that the local model's predictions meet the label distribution constraints while remaining clear and transparent.

In addition to designing the LIMEFLDL model specifically for the interpretability issues of LDL algorithms, we have also conducted proof analysis and experimental validation of the related theories of this model. First, we have provided analytical solution formulas for the improved interpretation algorithm under certain assumption conditions and proved the boundary conditions for the explanations to reach convergence and stability. Second, we have analyzed the properties of the interpretation algorithm, including local accuracy, dummy features, etc., and provided proofs, which enhances the theoretical completeness of the algorithm and also increases its usability and trustworthiness. Third, most traditional fidelity measures are only applicable to single-label cases. However, for label distribution scenarios, these measures cannot be directly applied. To address this, we have introduced metrics for LDL as fidelity measures. These measures can quantify the fidelity of the explanation model's output to the actual model's decisions. Finally, through numerical experiments and human experiments, we have verified that the proposed method offers high fidelity, consistency, and trustworthiness in the interpretation of LDL models.

Our main contributions can be summarized as follows.

- We propose the LIMEFLDL method, which is suitable for interpretability tasks of LDL algorithms. This method takes into account the label distribution in the local region, generating local linear models to approximate the overall behavior of the black-box model's output label distribution. It can be widely applied to interpret complex models in various LDL tasks.

- We conduct extensive theoretical proofs and property analyses of the proposed LIMEFLDL method, ensuring that the interpretations it provides in LDL tasks are reliable and effectively approximate the true decision-making process. We also validate the reliability of the proposed method through various experiments.

## 2. Related Works

**Label distribution learning.** LDL aims to learn a predictive function that maps features to the sample's label distribution, where each value in the distribution represents the degree to which the corresponding label describes the sample. (Geng, 2016) first introduced maximum entropy model to represent the prediction function, laying the foundation for LDL. Subsequently, many studies have extended traditional learning algorithms to adapt to LDL tasks. For example, (Geng & Hou, 2015) proposed LDSVR by extending the support vector machine to handle LDL problem. (Xing et al., 2016) applied the LogitBoost model to LDL tasks. (Shen et al., 2017) developed LDLF by extending the drivable decision tree model for LDL. To further improve the generalization ability of label distribution learning models, some research has focused on mining the correlations between labels, aiming to capture the interdependencies among labels (Jia et al., 2018; Zhao & Zhou, 2018; Zheng et al., 2018). In addition, there are also efforts that enhance the expressive power of models by mapping labels to a low-dimensional space through label embedding methods (Peng et al., 2018; Wang & Geng, 2018; Xu et al., 2019). Although these approaches have significantly advanced the performance of the LDL model, the integration of LDL with interpretability techniques remains a largely unexplored area, presenting opportunities for future research.

**Improvements to LIME.** LIME is one of the most widely used techniques for interpreting black-box models, and many studies have also made improvements to it. For example, (Shi et al., 2020) proposed a modification to the perturbation sampling method to handle the correlations between features, thereby improving the accuracy of the explanations. Like other explanation methods, LIME relies on reference points (also known as baseline inputs) to generate samples. In (Kapishnikov et al., 2019), both black and white

references are used. In contrast, (Fong & Vedaldi, 2017) employed a combination of constant, noisy, and gaussian blur references. DLIME (Zafar & Khan, 2019), ALIME (Shankaranarayana & Runje, 2019) and GLIME (Tan et al., 2023) applied auto-encoders, deterministic methods, and integration of the locality weighting function, respectively, into the sampling process methods to refine the weighting function, improving the accuracy and stability of the local surrogate model. It is worth mentioning that there are also some works that apply LIME to multi-label classification tasks (Kakogeorgiou & Karantzalos, 2021; Belal et al., 2023; Vinogradova & Myers, 2023). These methods directly decompose the multi-label classification task into a series of binary classification tasks and then use LIME to seek explanations for each of them. After reviewing the existing research related to LIME, there has been no direct work on interpretability for LDL tasks. This paper will attempt to address this issue.

## 3. Proposed Interpretability Algorithm

### 3.1. Problem formation

Let $\mathcal{X} = \{x_1, x_2, \ldots, x_n\}$, where $\mathcal{X} \in \mathbb{R}^{n \times f}$ represents the input space, $x_i$ denotes the $i$-th instance, $n$ is the number of instances, and $f$ is the feature dimension. $\mathcal{D} = \{\mathcal{D}_1, \mathcal{D}_2, \ldots, \mathcal{D}_n\}$, where $\mathcal{D} \in \mathbb{R}^{n \times r}$ represents the output space, and $r$ is the number of labels. Each $\mathcal{D}_i = \left[d_{x_i}^{y_1}, d_{x_i}^{y_2}, \ldots, d_{x_i}^{y_r}\right]$ is the label distribution for $x_i$, where $d_{x_i}^{y_j}$ represents the description degree of the label $y_j$ in the label distribution of instance $x_i$, satisfying $d_{x_i}^{y_j} \in [0, 1]$ and $\sum_{j=1}^{r} d_{x_i}^{y_j} = 1$. The goal of LDL is to learn a mapping function $\varsigma$ from $\mathcal{X}$ to $\mathcal{D}$ that can predict the label distributions for unseen instances. Furthermore, in order to interpretate the LDL model, we aim to approximate the local behavior of the complex LDL mapping $\varsigma$ by simple linear model, i.e., $\min \|\varsigma_{\text{simple}} - \varsigma\|_F^2$, where $\varsigma_{\text{simple}}$ is the simple linear model.

### 3.2. Main framework

The main goal of our work is to train a local linear model to approximate the overall behavior of a black-box model in LDL. For a given instance $x_i$ that needs to be explained, we generate $m$ local sampling instances for it and transform them into an interpretable feature representation matrix $\boldsymbol{Z} = (z_{jq}) \in \{0, 1\}^{m \times f}$, where $z_{jq}$ represents the $q$-th interpretable feature representation of the $j$-th local sampling instance. If the instance $x_i$ is an image, we can use a segmentation algorithms, such as Quickshift (Vedaldi & Soatto, 2008), to divide it into $f$ blocks. For the generated $j$-th local sampling instance, $z_{jq} = 1$ indicates the $q$-th block using the original image block, and $z_{jq} = 0$ indicates the $q$-th block using the average of the channels. If $x_i$ is an instance from tabular data, each column is divided into

several individual boxes based on entropy algorithm (Garreau & von Luxburg, 2022). For the generated $j$-th local sampling instance, $z_{jq} = 1$ indicates that the $q$-th feature is in the same box as the value of the $q$-th feature of $x_i$, and $z_{jq} = 0$ indicates that the $q$-th feature is in a different box as the value of the $q$-th feature of $x_i$. In summary, for the to-be-explained instance $x_i$, we generate the corresponding local sampling instance matrix $\boldsymbol{Z}$ based on the Bernoulli distribution.

Additionally, since each feature in LDL corresponds to multiple labels, we generate an attribution vector $a_i$ for each feature, the length of $a_i$ corresponds to the labeling sequence, with $a_i \in \mathbb{R}^{r \times 1}$, where each element $a_{ij}$ denotes the attribution value of the $i$-th feature to the $j$-th label. These feature attribution values are then combined to form the feature attribution matrix $\boldsymbol{A}$. We uses uniform initialization as initialization. The local linear model for LDL can thus be expressed as follows:

$$G\left(\boldsymbol{Z}\right) = \boldsymbol{Z}\boldsymbol{A}^\top + \boldsymbol{b}, \tag{1}$$

where $\boldsymbol{Z} \in \{0, 1\}^{m \times f}$ is the distribution matrix sampled from the instances to be interpreted, $\boldsymbol{A} \in \mathbb{R}^{r \times f}$ is the feature attribution matrix, and $\boldsymbol{b} \in \mathbb{R}^{r \times 1}$ is the bias term. We can simplify the equation as follows:

$$G\left(\boldsymbol{Z}\right) = \left[\boldsymbol{Z}; \boldsymbol{1}\right]\left[\boldsymbol{A}^\top, \boldsymbol{b}^\top\right] = \boldsymbol{Z}'\boldsymbol{A}'^\top. \tag{2}$$

However, each locally sampled instance differs from the one to be interpreted, for each instance $\boldsymbol{z_i}$ in $\boldsymbol{Z}$, we assign a sampling weight $\pi(\boldsymbol{z_i})$, and the sampling weight matrix $\boldsymbol{\Pi}$ is then defined for the entire locally sampled set $\boldsymbol{Z}$.

$$\pi(\boldsymbol{z_i}) = \exp\left(-\sigma^{-2}\|\boldsymbol{1} - \boldsymbol{z_i}\|_2^2\right), \boldsymbol{\Pi} = \text{diag}(\pi(\boldsymbol{z_i})), \tag{3}$$

where $\sigma$ is the kernel width parameter, and $\boldsymbol{1}$ denotes the all-one vector as the interpretable representation of the term to be interpreted. In this framework, we use the $L_2$ norm by default. However, other metrics such as the $L_1$ norm, cosine, or euclidean distance can also be applied. Typically, cosine distance is used for image data, while the $L_1$ norm is preferred for tabular data. For additional analysis, please refer to the Appendix A.3.

We restore the logical values in $\boldsymbol{Z}$ back to the corresponding original input $h(\boldsymbol{Z})$ through a process of inverse discretization, and then input into the original black-box model $F$ to obtain the label distribution on locally sampled instances, $F(h(\boldsymbol{Z}))$. To ensure interpretability and fidelity, our objective is to minimize the distance between $G\left(\boldsymbol{Z}\right)$ and $F(h(\boldsymbol{Z}))$, while still satisfying the LDL constraints. Since not all interpretations of $\boldsymbol{A}'$ are simple and smooth, we use $\|\boldsymbol{A}'\|_F^2$ to constrain the complexity of the interpretation. We use the alternating direction method of multipliers (ADMM) (Boyd et al., 2011) to solve objective functions with linear constraints. Ultimately, we express the LIMEFLDL

generation as an explanation.

$$\min_{\boldsymbol{A}'} \frac{1}{2m} \left\| \boldsymbol{\Pi}(\boldsymbol{Z}'\boldsymbol{A}'^\top - F(h(\boldsymbol{Z}))) \right\|_F^2 + \frac{\lambda}{2m} \|\boldsymbol{A}'\|_F^2, \quad (4)$$
$$\text{s.t.} \quad \boldsymbol{Z}'\boldsymbol{A}'^\top \times \mathbf{1}_{r \times 1} = \mathbf{1}_{m \times 1}, \boldsymbol{Z}'\boldsymbol{A}'^\top \geq \mathbf{0}_{m \times r}.$$

Equation (4) can be solved by the following alternative methods in iteration $t$:

$$\boldsymbol{A}'^\top_{t+1} = \arg\min_{\boldsymbol{A}'^\top_t} \frac{1}{2m} \left\| \boldsymbol{\Pi}(\boldsymbol{Z}'\boldsymbol{A}'^\top_t - F(h(\boldsymbol{Z})) \right\|_F^2$$
$$+ \frac{\rho}{2m} \left\| \boldsymbol{Z}'\boldsymbol{A}'^\top_t \times \mathbf{1}_{r \times 1} - \mathbf{1}_{m \times 1} \right\|_2^2 + \frac{\lambda}{2m} \|\boldsymbol{A}'^\top_t\|_F^2$$
$$+ \left\langle \frac{\boldsymbol{u}^{(t)}}{m}, \boldsymbol{Z}'\boldsymbol{A}'^\top_t \times \mathbf{1}_{r \times 1} - \mathbf{1}_{m \times 1} \right\rangle$$
$$+ \frac{v}{2m} \sum \max(-\boldsymbol{Z}'\boldsymbol{A}'^\top_t, 0), \quad (5)$$
$$\boldsymbol{u}^{(t+1)} = \boldsymbol{u}^{(t)} + \frac{\rho}{m} \left( \boldsymbol{Z}'\boldsymbol{A}'^\top_{t+1} \times \mathbf{1}_{r \times 1} - \mathbf{1}_{m \times 1} \right), \quad (6)$$

where $m$ represents the number of sampled instances, $\boldsymbol{u} \in \mathbb{R}^{m \times 1}$ is the Lagrange multiplier, and $\rho$ and $v$ are the penalty factors. For the non-negative constraint, we introduce a penalty term and provide Assumption 3.1 to ensure that the non-negative constraint is satisfied.

**Assumption 3.1.** For a given $\epsilon > 0$, $\exists N > 0$, $\frac{v}{m} \sum \max(-\boldsymbol{Z}'\boldsymbol{A}'^\top_t, 0) < \epsilon$ holds for any $\frac{v}{m} > N$, we assume that the nonnegative constraint is satisfied.

Here, we use the limited-memory quasi-Newton method (L-BFGS) for optimization. The computation of L-BFGS is mainly related to the first-order gradient, which can be obtained by:

$$\nabla \boldsymbol{A}'^\top_{t+1} = \boldsymbol{Z}'^\top \boldsymbol{\Pi} \left( \boldsymbol{Z}'\boldsymbol{A}'^\top_t - F(h(\boldsymbol{Z})) \right) + \boldsymbol{Z}'^\top \boldsymbol{u} \mathbf{1}^\top_{r \times 1}$$
$$+ \boldsymbol{Z}'^\top \rho \left( \boldsymbol{Z}'\boldsymbol{A}'^\top_t \mathbf{1}_{r \times 1} - \mathbf{1}_{m \times 1} \right) \mathbf{1}_{r \times 1}. \quad (7)$$

To ensure that the final interpretation model is concise and efficient, we must select the most important features. Therefore, we design a new scoring function $S$, which accurately evaluates the contribution of each feature to the overall distribution of the labels.

$$S = 1 - \frac{\mathrm{KL}\left( G\left(\boldsymbol{Z}_c\right), F((h(\boldsymbol{Z})) \right)}{\mathrm{KL}\left( \boldsymbol{Y}_{\text{mean}}, F((h(\boldsymbol{Z})) \right)}, \quad (8)$$

where $F(h(\boldsymbol{Z}))$ represents the output of the black-box model, and $G\left(\boldsymbol{Z}_c\right)$ denotes the output of the local linear model based on the selected features. $\boldsymbol{Y}_{\text{mean}}$ is the uniform distribution. KL divergence are used to measure the distance between these two outputs. Based on the value ranking of $S$, we select the most explanatory features.

### 3.3. Analytical solution analysis

To better understand the specific form of the feature attribution matrix $\boldsymbol{A}'$ when the constraints are minimized, we first provide a simplified form. All proofs for theorems and properties presented in this section can be found in Appendix B.

$$\boldsymbol{Z}'^\top \boldsymbol{\Pi} \boldsymbol{Z}' = \boldsymbol{B}, \qquad \boldsymbol{Z}'^\top \boldsymbol{\Pi} F(h(\boldsymbol{Z})) = \boldsymbol{D},$$
$$\rho \boldsymbol{Z}'^\top \boldsymbol{Z}' = \boldsymbol{C}, \qquad \boldsymbol{Z}'^\top \boldsymbol{u} \mathbf{1}_{1 \times r} = \boldsymbol{L},$$
$$\mathbf{1}_{r \times 1} \times \mathbf{1}_{1 \times r} = \boldsymbol{E}, \qquad \rho \boldsymbol{Z}'^\top \mathbf{1}_{m \times 1} \mathbf{1}_{1 \times r} = \boldsymbol{O},$$

$$\boldsymbol{B} = \frac{1}{m}(\sum_i \pi(\boldsymbol{z_i})((z_{ij})^2 - (z_{ij}z_{iq}))\boldsymbol{I} + \sum_i \pi(\boldsymbol{z_i})(z_{ij}z_{iq})\mathbf{1}\mathbf{1}^\top). \quad (9)$$

Set the gradient to zero to obtain the iterative form of $\boldsymbol{A}'$,

$$\boldsymbol{A}'^\top = \boldsymbol{B}^{-1} \left( \boldsymbol{D} - \boldsymbol{L} + \boldsymbol{O} - \boldsymbol{C}\boldsymbol{A}'^\top \boldsymbol{E} \right), \quad (10)$$

where $\|\boldsymbol{B}^{-1}\|_F \leq (f+1)^{\frac{1}{2}} 2^f e^{\frac{1}{\sigma^2}}$, $\|\boldsymbol{C}\|_F \in \left[0, f^2 \rho m^{-1}\right]$, $\|\boldsymbol{E}\|_F = r/m$. From this we give an iterative form for $\boldsymbol{A}'$.

**Theorem 3.2.** *If Assumption 3.1 holds and $\boldsymbol{Z}'$ follows the $\{0,1\}$ distribution, ignoring the regularization constraints, the condition for convergence is given by: $\rho < 2^{-f}(f+1)^{-\frac{1}{2}} m^2 f^{-2} r^{-1} e^{-\frac{1}{\sigma^2}}$. By the Contraction Mapping Principle (Banach fixed-point principle), there exists an iterative fixed point $\boldsymbol{A}'^*$ that minimizes the loss.*

However, Theorem 3.2 only shows that the feature attribution matrix has iterative fixed points, but it does not provide a specific form due to the iterative structure. We find that this is caused by the $L_2$ norm constraints on vectors with a sum of 1. Building Theorem 3.2, to derive an approximate analytic solution, we apply linear constraints using the lagrange multiplier method and the KKT conditions. This allows us to obtain an analytic solution for $\boldsymbol{A}'$.

$$\boldsymbol{A}'^\top = \left( \boldsymbol{Z}'^\top \boldsymbol{\Pi} \boldsymbol{Z}' \right)^{-1} \left( \boldsymbol{Z}'^\top \boldsymbol{\Pi} F(h(\boldsymbol{Z})) - \boldsymbol{Z}'^\top \boldsymbol{u} \mathbf{1}_{1 \times r} \right),$$
$$\text{s.t.} \quad \boldsymbol{Z}'\boldsymbol{A}'^\top \times \mathbf{1}_{r \times 1} = \mathbf{1}_{m \times 1}. \quad (11)$$

We divide the solution into two parts: $\boldsymbol{Z}'^\top \boldsymbol{\Pi} \boldsymbol{Z}' = \boldsymbol{\Delta}$, $\boldsymbol{Z}'^\top \boldsymbol{\Pi} F(h(\boldsymbol{Z})) - \boldsymbol{Z}'^\top \boldsymbol{u} \mathbf{1}_{1 \times r} = \boldsymbol{\Gamma}$, which allows us to determine the constraint ranges for both the numerical and desired solutions of $\boldsymbol{A}'$.

$$\left\| \boldsymbol{\Delta_m}^{-1} \boldsymbol{\Gamma_m} - \widetilde{\boldsymbol{\Delta}}^{-1} \widetilde{\boldsymbol{\Gamma}} \right\|_F \leq \left\| \widetilde{\boldsymbol{\Delta}}^{-1} \right\|_F \left\| \boldsymbol{\Gamma_m} - \widetilde{\boldsymbol{\Gamma}} \right\|_F + \left\| \boldsymbol{\Delta_m}^{-1} - \widetilde{\boldsymbol{\Delta}}^{-1} \right\|_F \left\| \boldsymbol{\Gamma_m} \right\|_F, \quad (12)$$

where $\boldsymbol{\Delta_m}$ represents the computed value of $\boldsymbol{\Delta}$ obtained through the model, $\widetilde{\boldsymbol{\Delta}}$ denotes the expected value. Similarly, $\boldsymbol{\Gamma_m}$ corresponds to the computed value of $\boldsymbol{\Gamma}$, $\widetilde{\boldsymbol{\Gamma}}$ denotes the expected value. $\boldsymbol{A}'^\top_m = \boldsymbol{\Delta_m}^{-1} \boldsymbol{\Gamma_m}$ represents the computed value of $\boldsymbol{A}'^\top$, $\widetilde{\boldsymbol{A}}'^\top$ denotes the expected value.

From Appendix B.1 and the form of $\boldsymbol{\Delta}$, we can conclude each element of $\boldsymbol{\Delta_m} - \widetilde{\boldsymbol{\Delta}}$ falls within the range $\left[-\frac{1}{4}, 1\right]$.

Using the matrix Hoeffding inequality (Tropp, 2015), for any $t > 0$, we have,

$$\mathbb{P}\left(\left\|\boldsymbol{\Delta}_{\boldsymbol{m}} - \widetilde{\boldsymbol{\Delta}}\right\|_F \geq t\right) \leq 2f \exp\left(-\frac{mt^2}{8f^2}\right). \quad (13)$$

Similarly, given $F(h(\boldsymbol{Z})) \subseteq [0,1]^{m \times r}$, $\pi(\cdot) \in [0,1]$, and $\boldsymbol{u} \subseteq (0,1)^{m \times 1}$, the form of $\boldsymbol{\Gamma}$ ensures that each element of $\boldsymbol{\Gamma}_{\boldsymbol{m}} - \widetilde{\boldsymbol{\Gamma}}$ is bounded by $[-1,1]$. Thus, for any $t > 0$, we have,

$$\mathbb{P}\left(\left\|\boldsymbol{\Gamma}_{\boldsymbol{m}} - \widetilde{\boldsymbol{\Gamma}}\right\|_F \geq t\right) \leq 2f \exp\left(-\frac{mt^2}{8f^2}\right), \quad (14)$$

where $\mathbb{P}$ represents the probability that the Frobenius norm of each matrix satisfies the given condition.

**Theorem 3.3 (Convergence).** *Based on Theorem 3.2 $\forall \epsilon > 0, \delta \in (0,1)$, if $m = \Omega(\max(2^{5+2f}(f + 1)f^3 e^{\frac{2}{\sigma^2}} \epsilon^{-2} \ln\left(\frac{4f}{\delta}\right), 2^5 f^{\frac{5}{2}} r^{\frac{1}{2}} \epsilon^{-2} \ln(\frac{4f}{\delta}))$, then we have $\mathbb{P}(\|\boldsymbol{A}_{\boldsymbol{m}}'^{\top} - \widetilde{\boldsymbol{A}}'^{\top}\|_F < \epsilon) \geq 1 - \delta$, and $\widetilde{\boldsymbol{A}}'^{\top} = \lim_{m \to \infty} \boldsymbol{A}_{\boldsymbol{m}}'^{\top}$, where $\Omega$ represents the lower bound on the number of instances required for this problem.*

We aim not only to ensure the solution converges, but also to ensure that minor changes to the black-box model do not affect the interpretation. That is the stability of the interpretation.

**Theorem 3.4 (Stability).** *Based on Theorem 3.3, for any two similar black-box models $P$ and $Q$, define $T(h(\boldsymbol{Z})) = |P(h(\boldsymbol{Z})) - Q(h(\boldsymbol{Z}))|$, and $T(h(\boldsymbol{Z}))$ of each element $t(h(\boldsymbol{Z})) < \epsilon$, there exists a constant $\lambda \in \left[0, mf2^{-\frac{1}{2}+f}r^{\frac{1}{2}}e^{\frac{1}{\sigma^2}}\epsilon\right]$ such that $\|\boldsymbol{A}_{\boldsymbol{P}}'^{\top} - \boldsymbol{A}_{\boldsymbol{Q}}'^{\top}\|_F \leq \lambda$.*

The mathematic expectation of the explanation is calculated according to the analytical solution formula as the number of sampling instances increases exponentially. The explanations obtained from the model are then compared, and the Top-20 Jaccard index is computed for both. As shown in Figure 1(a), as the number of sampling instances grows, the two explanations become more similar and eventually converge. This confirms the correctness of convergence. However, due to the inherent rounding errors in the numerical computation of the inverse matrix, the calculation of $\boldsymbol{A}^{-1}$ may instead result in $(\boldsymbol{A} + \delta\boldsymbol{A})^{-1}$, where $\delta$ is a small error matrix. This introduces inaccuracies, affecting the final index results. As shown in Figure 1(b), the difference gradually decreases, and the interpretations of the two black-box model outputs become increasingly consistent. This demonstrates the correctness of stability. Therefore, we conclude that our explanation algorithm is stable in LDL task and can effectively approximate the real decision-making process. The overall flow of our algorithm is outlined in Algorithm 1.

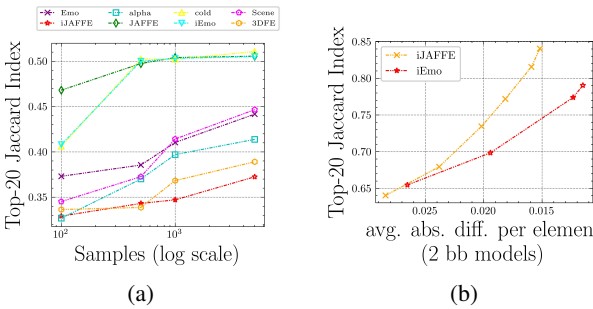

*Figure 1.* Convergence and stability visualization. (a): Convergence visualization, where x-axis denotes the exponentially increasing number of samples (log scale), and y-axis denotes the Top 20 Jaccard Index. It demonstrates convergence as Jaccard Index ↑ and stabilized. (b): Stability visualization, where x-axis denotes the decreasing divergence between two black-box models, and y-axis denotes the Top 20 Jaccard Index. It shows Jaccard Index ↑ when predictive distribution divergence between black-box models ↓ (x-axis).

---

**Algorithm 1** LIMEFLDL

---

**Input**: instance to be interpreted $x$, LDL model $F$, sampling sample size $m$, weight kernel $\pi$.
**Initialization**: $\boldsymbol{A_0}$, $\boldsymbol{u^{(0)}}$, $\rho$, $t = 1$.
**Output**: $\boldsymbol{A}$.
1: $\boldsymbol{\Pi}, \boldsymbol{Z} \leftarrow \{\}$;
2: **for** $i \in \{1, 2, \ldots, m\}$ **do**
3: $\quad z_i \leftarrow sample\_around\{x\}$;
4: $\quad \pi(z_i) \leftarrow z_i$; $\qquad\qquad\qquad \triangleright$ Equation (3)
5: $\quad \boldsymbol{Z} \leftarrow \boldsymbol{Z} \cup (z_i), \boldsymbol{\Pi} \leftarrow \pi(z_i)$;
6: **end for**
7: **while** the stopping criterion is not satisfied **do**
8: $\quad \boldsymbol{Z_c} \leftarrow S(\boldsymbol{Z})$; $\qquad\qquad\quad \triangleright$ Equation (8)
9: $\quad$ update $\boldsymbol{A}_{t+1}$; $\qquad\qquad\quad \triangleright$ Equation (5)
10: $\quad$ update $\boldsymbol{u}^{(t+1)}$; $\qquad\qquad\quad \triangleright$ Equation (6)
11: $\quad t \leftarrow t + 1$;
12: **end while**
13: **return** $\boldsymbol{A}$;

---

### 3.4. Analysis of algorithm properties

To make the interpretation algorithm easy to understand, we highlight some important properties that aid in its use. First, we show that the explanatory model is consistent with additive feature attribution (Lundberg & Lee, 2017). We present Property 3.5 to illustrate this.

**Property 3.5.** Let $\boldsymbol{x}$ be the instance to be interpreted, and $\boldsymbol{\xi}$ be its interpretable data representation. $\xi_i$ represents the $i$-th interpretable feature representation. Let $F$ denote the black-box model and $G$ represent the locally interpretable model. Then, we have

$$F(\boldsymbol{x}) \approx G(\boldsymbol{\xi}) = \phi_{\boldsymbol{0}} + \left[\sum_{i=1}^{f} \xi_i a_{i1} \cdots \sum_{i=1}^{f} \xi_i a_{ir}\right]^{\top}, \quad (15)$$

where $\phi_{\boldsymbol{0}} = F(h(\boldsymbol{0}))$ is the output corresponding to an

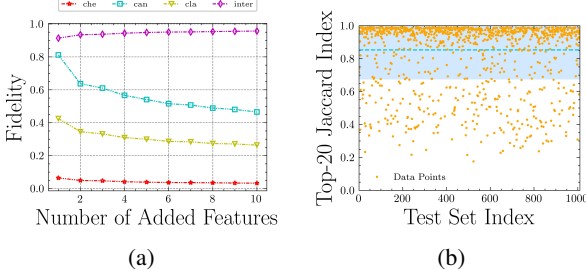

*Figure 2.* Local accuracy and dummy features visualization. (a): Local accuracy in multiple forms of fidelity, where x-axis denotes the number of added features, and y-axis denotes the forms of fidelity. (b): Dummy features, where x-axis denotes the index of test set example, and y-axis denotes the top-20 Jaccard indices.

all-0 vector. This represents the output when no features are selected, which is consistent with the architecture of additive feature attribution.

**Property 3.6.** For any interpretable representation $\boldsymbol{\xi}$, let its $k$-th feature be $\xi_k^i$ or $\xi_k^j$, where $\xi_k^i \neq \xi_k^j$. If the interpretation model $G(\xi_k^i, \xi_{f\setminus k}) = G(\xi_k^j, \xi_{f\setminus k})$, then the feature attribution of the $k$-th is 0, indicating that the feature is dummy.

**Property 3.7.** For any interpretable representation $\boldsymbol{\xi}$, let its $i$-th and $j$-th features be $\xi_i$ and $\xi_j$, where $\xi_i \neq \xi_j$. If these two feature values are swapped to form $\boldsymbol{\xi}'$, and the explanatory model satisfies $G(\boldsymbol{\xi}) = G(\boldsymbol{\xi}')$, then the feature attribution for the $i$-th and $j$-th features remains the same.

Some features contribute positively or negatively to the model's output, while others do not. To empirically validate these observations. Initially, we first apply a mask to the image based on the mean value of each superpixel block. This mask has impact on the interpretation model's predictions. Next, the highest-ranked features are gradually added. As shown in Figure 2(a), as the number of features increases, the fidelity of the local model improves consistently, indicating that some features contribute positively or negatively to the model's output and the local accuracy is both reliable and precise. To demonstrate that certain features do not affect model outputs, for each test image, the lowest-ranked feature is marked with a black smudge and outlined in green. Clean and processed images are evaluated under the same black-box model. Top-20 Jaccard indices are computed, with a dashed blue line showing the mean and a shaded area for standard deviation. As shown in Figure 2(b) explanations remain unchanged for some images, exhibit minor variations for others, and undergo significant changes in a small subset.

# 4. Experiments

In this section, we will evaluate the proposed method on eight real-world datasets, test the fidelity and robustness of the explanation algorithm, and conduct human experiments to assess whether meaningful explanations are provided to end-users.

## 4.1. Numerical data sets

We conduct experiments on eight label distribution datasets from different kinds of real-world tasks to evaluate our approach. Yeast_alpha (alpha) and Yeast_cold (cold) (Geng, 2016) are collected from biological experiments. JAFFE (Lyons et al., 1998), SBU_3DFE (3DFE) (Yin et al., 2006), and Emotion6 (Emo) (Peng et al., 2015) are collected from sentiment mining tasks. Natural Scene(Scene) (Geng, 2016) is generated from a natural scene recognition task. JAFFE and Emotion6 have both image data and processed tabular data. For the tabular data in these six datasets, we employ table-based interpretation approaches. For the JAFFE (iJAFFE) and Emotion6 (iEmo) image datasets, we apply image interpretation methods directly on visual data. The details of all eight data sets are summarized in Table 1.

*Table 1.* Statistics from eight real-world datasets.

| ID | Dataset | # examples | # features | # labels |
|---|---|---|---|---|
| 1 | Yeast_alpha(alpha) | 2465 | 24 | 18 |
| 2 | Yeast_cold(cold) | 2465 | 24 | 4 |
| 3 | JAFFE | 213 | 243 | 6 |
| 4 | SBU_3DFE (3DFE) | 2500 | 243 | 6 |
| 5 | Emotion6 (Emo) | 1980 | 168 | 7 |
| 6 | Natural Scene (Scene) | 2000 | 294 | 9 |
| 7 | Image-JAFFE (iJAFFE) | 213 | 243 | 6 |
| 8 | Image-Emotion6 (iEmo) | 1980 | 168 | 7 |

## 4.2. Numerical experimental evaluations

Fidelity is one of the most widely used metrics in interpretable artificial intelligence to evaluate how well an interpretation model mimics black-box decisions. Depending on the type of interpretation, as analyzed in (Guidotti et al., 2019), fidelity can be specialized in various forms. In our setup, we evaluate the fidelity using six distinct metrics, grouped into two categories. The first group includes Canberra distance, Clark distance, Kullback-Leibler divergence, and Chebyshev distance metrics, which measure the distance between two distributions. Lower values ($\downarrow$) of these metrics indicate better fidelity. The second group comprises intersection similarity and cosine similarity metrics, which assess the similarity between distributions. Higher values ($\uparrow$) of these metrics represent better performance. For the

*Table 2.* The effectiveness of two interpretation algorithms is evaluated on different datasets, using fidelity and consistency metrics under black-box modeling with RBF-LDL-LRR. Results are marked with ● for better outcomes and ○ for anomalous data.

| Index | Algorithms | Chebyshev ↓ | Clark ↓ | Canberra ↓ | KL ↓ | Cosine ↑ | Intersection ↑ | Jaccard ↑ |
|---|---|---|---|---|---|---|---|---|
| 1 | PULIME | $.0009 \pm_{.0000}$ | $.0137 \pm_{.0001}$ | $.0452 \pm_{.0002}$ | $.0001 \pm_{.0000}$ | $.9999 \pm_{.0000}$ | $.9975 \pm_{.0000}$ | ●$.6428 \pm_{.3192}$ |
| 1 | LIMEFLDL | ●$.0006 \pm_{.0000}$ | ●$.0108 \pm_{.0000}$ | ●$.0358 \pm_{.0003}$ | ●$.0000 \pm_{.0000}$ | ●$1.000 \pm_{.0000}$ | ●$.9980 \pm_{.0000}$ | $.5044 \pm_{.3426}$ |
| 2 | PULIME | $.0035 \pm_{.0000}$ | ●$.0089 \pm_{.0001}$ | ●$.0153 \pm_{.0002}$ | ○$-.0001 \pm_{.0001}$ | ●$.9999 \pm_{.0000}$ | ●$.9961 \pm_{.0001}$ | $.3608 \pm_{.2175}$ |
| 2 | LIMEFLDL | ●$.0033 \pm_{.0000}$ | $.0096 \pm_{.0000}$ | $.0165 \pm_{.0001}$ | ●$.0001 \pm_{.0000}$ | $.9999 \pm_{.0000}$ | $.9960 \pm_{.0000}$ | ●$.6200 \pm_{.2843}$ |
| 3 | PULIME | $.0461 \pm_{.0029}$ | $.1990 \pm_{.0107}$ | $.3763 \pm_{.0210}$ | ○$-.0306 \pm_{.0050}$ | $.9903 \pm_{.0011}$ | $.9403 \pm_{.0036}$ | $.5790 \pm_{.0957}$ |
| 3 | LIMEFLDL | ●$.0338 \pm_{.0018}$ | ●$.1493 \pm_{.0074}$ | ●$.2949 \pm_{.0127}$ | ●$.0078 \pm_{.0009}$ | ●$.9935 \pm_{.0006}$ | ●$.9538 \pm_{.0020}$ | ●$.7080 \pm_{.0753}$ |
| 4 | PULIME | $.0583 \pm_{.0008}$ | $.1916 \pm_{.0020}$ | $.3601 \pm_{.0033}$ | ○$-.0320 \pm_{.0013}$ | $.9870 \pm_{.0003}$ | $.9361 \pm_{.0007}$ | $.4318 \pm_{.0707}$ |
| 4 | LIMEFLDL | ●$.0348 \pm_{.0005}$ | ●$.1328 \pm_{.0017}$ | ●$.2613 \pm_{.0034}$ | ●$.0075 \pm_{.0002}$ | ●$.9926 \pm_{.0002}$ | ●$.9559 \pm_{.0006}$ | ●$.4675 \pm_{.0540}$ |
| 5 | PULIME | ●$.0001 \pm_{.0000}$ | ●$.0001 \pm_{.0000}$ | ●$.0001 \pm_{.0000}$ | ○$-.0001 \pm_{.0000}$ | $1.000 \pm_{.0000}$ | ●$.9999 \pm_{.0000}$ | $.1556 \pm_{.1245}$ |
| 5 | LIMEFLDL | $.0001 \pm_{.0001}$ | $.0749 \pm_{.0027}$ | $.0782 \pm_{.0028}$ | ●$.0001 \pm_{.0000}$ | $1.000 \pm_{.0000}$ | $.9999 \pm_{.0000}$ | ●$.3804 \pm_{.0468}$ |
| 6 | PULIME | $.0010 \pm_{.0000}$ | ●$.0151 \pm_{.0002}$ | ●$.0398 \pm_{.0005}$ | $.0011 \pm_{.0001}$ | $.9997 \pm_{.0000}$ | $.9986 \pm_{.0000}$ | $.2846 \pm_{.3556}$ |
| 6 | LIMEFLDL | ●$.0008 \pm_{.0000}$ | $.0154 \pm_{.0001}$ | $.0395 \pm_{.0006}$ | ●$.0007 \pm_{.0001}$ | ●$.9999 \pm_{.0000}$ | ●$.9989 \pm_{.0001}$ | ●$.3487 \pm_{.2426}$ |
| 7 | PULIME | $.0040 \pm_{.0000}$ | $.0142 \pm_{.0002}$ | $.0289 \pm_{.0005}$ | $.0002 \pm_{.0001}$ | $.9998 \pm_{.0000}$ | $.9951 \pm_{.0001}$ | $.7470 \pm_{.1220}$ |
| 7 | LIMEFLDL | ●$.0033 \pm_{.0000}$ | ●$.0116 \pm_{.0002}$ | ●$.0236 \pm_{.0005}$ | ●$.0000 \pm_{.0000}$ | ●$.9999 \pm_{.0000}$ | ●$.9960 \pm_{.0000}$ | ●$.7903 \pm_{.1054}$ |
| 8 | PULIME | ●$.0003 \pm_{.0000}$ | ●$.0032 \pm_{.0001}$ | ●$.0069 \pm_{.0002}$ | ○$-.0001 \pm_{.0000}$ | $.9999 \pm_{.0000}$ | ●$.9996 \pm_{.0000}$ | $.1591 \pm_{.1298}$ |
| 8 | LIMEFLDL | $.0033 \pm_{.0000}$ | $.0116 \pm_{.0002}$ | $.0236 \pm_{.0005}$ | ●$.0001 \pm_{.0000}$ | $.9999 \pm_{.0000}$ | $.9960 \pm_{.0001}$ | ●$.7903 \pm_{.1054}$ |

instances to be explained, we calculate the distance between the label distributions predicted by the black-box model and those output by the local linear model. This serves as the fidelity metric for the interpretation model.

Consistency is a fundamental requirement for any trustworthy method. Here, we compute the average top-K Jaccard index (JI) for interpretations generated using 10 different random seeds. Let $w_1, \ldots, w_{10}$ represent the interpretations obtained from these seeds. For the first $K$ highest-ranked features in $w_i$, the set is denoted as $\mathcal{R}_{i,K}$. We then calculate the average Jaccard index between $\mathcal{R}_{i,K}$ and $\mathcal{R}_{j,K}$, defined as: $\text{JI}(\mathcal{R}_{i,K}, \mathcal{R}_{j,K}) = \frac{|\mathcal{R}_{i,K} \cap \mathcal{R}_{j,K}|}{|\mathcal{R}_{i,K} \cup \mathcal{R}_{j,K}|}$. A JI value closer to 1 indicates greater stability in interpretation consistency. The instances used in this analysis are randomly selected from the test set, ensuring diverse coverage across random seeds.

### 4.3. Experimental setup for numerical experiments

The proposed interpretation algorithm, in conjunction with the parallel use of the traditional LIME algorithm (PULIME), is applied to widely used LDL methods, including LDL-SCL (Zheng et al., 2018), LDL-LRR (Jia et al., 2023), AA-KNN and the Maximum Entropy Model (MEM). Since the feature extraction module in the LDL-LRR model is relatively simple, we incorporate a Gaussian kernel function to enhance feature extraction during training (RBF-LDL-LRR). This improves the model's performance while also increasing its complexity. All codes are provided by the original authors and we adopt the recommended parameters reported in their respective studies. For image datasets, using the $L_2$ norm as a distance metric during local sampling

proved problematic. Apart from the target sample, all other local samples received a weight of zero. To address this, we use cosine distance for local sampling in image datasets. This approach ensures that each local sample is assigned a weight proportional to its distance from the target sample. For tabular datasets, we retain the $L_2$ norm as a distance metric. Detailed parameter configurations for LIMEFLDL and PULIME are documented in the Appendix A.1. These settings align with the implementations used in our numerical experiments.

### 4.4. Human experiments

In addition to the numerical experiments, we conduct human interpretability experiments to evaluate whether LIMEFLDL offers more meaningful interpretations for users. The experiment has two parts, with 10 participants in each. Details of the experimental process are provided below.

*Can LIMEFLDL improve the understanding of model predictions?* To evaluate this, we select images that the black-box model predicted correctly. The participants are shown both the original image and the generated interpretation. They are then asked to assess the degree of agreement between the interpretation and their intuitive understanding of each label. For each label, the participants first evaluate whether the provided reason is reasonable. If so, they then rate the degree of agreement on a scale of 1 to 5, where 1 indicates no match and 5 indicates perfect agreement.

*Can LIMEFLDL help identify modeling errors?* To exploit this, we select images where the black-box model showed a significant difference between the predicted and actual label

*Table 3.* The effectiveness of two interpretation algorithms is evaluated on different datasets, using fidelity and consistency metrics under black-box modeling with LDL-SCL. Results are marked with ● for better outcomes and ○ for anomalous data.

| Index | Algorithms | Chebyshev ↓ | Clark ↓ | Canberra ↓ | KL ↓ | Cosine ↑ | Intersection ↑ | Jaccard ↑ |
|---|---|---|---|---|---|---|---|---|
| 1 | PULIME | ●$.0008\pm_{.0000}$ | ●$.0131\pm_{.0001}$ | ●$.0431\pm_{.0003}$ | ○$-.0001\pm_{.0000}$ | ●$.9999\pm_{.0000}$ | ●$.9976\pm_{.0000}$ | ●$.6446\pm_{.1244}$ |
| 1 | LIMEFLDL | $.0009\pm_{.0000}$ | $.0150\pm_{.0002}$ | $.0501\pm_{.0007}$ | ●$.0000\pm_{.0000}$ | $.9999\pm_{.0000}$ | $.9972\pm_{.0000}$ | $.6434\pm_{.1732}$ |
| 2 | PULIME | ●$.0034\pm_{.0000}$ | ●$.0087\pm_{.0001}$ | ●$.0149\pm_{.0002}$ | ○$-.0002\pm_{.0001}$ | ●$.9999\pm_{.0000}$ | ●$.9962\pm_{.0001}$ | $.3924\pm_{.0956}$ |
| 2 | LIMEFLDL | $.0037\pm_{.0000}$ | $.0097\pm_{.0001}$ | $.0168\pm_{.0002}$ | ●$.0001\pm_{.0000}$ | $.9999\pm_{.0000}$ | $.9958\pm_{.0001}$ | ●$.7558\pm_{.1823}$ |
| 3 | PULIME | $.0523\pm_{.0022}$ | $.2236\pm_{.0091}$ | $.4222\pm_{.0191}$ | ○$-.0368\pm_{.0088}$ | $.9885\pm_{.0009}$ | $.9332\pm_{.0031}$ | $.5687\pm_{.0824}$ |
| 3 | LIMEFLDL | ●$.0337\pm_{.0018}$ | ●$.1468\pm_{.0088}$ | ●$.2903\pm_{.0170}$ | ●$.0077\pm_{.0008}$ | ●$.9936\pm_{.0006}$ | ●$.9544\pm_{.0025}$ | ●$.6508\pm_{.0880}$ |
| 4 | PULIME | $.0326\pm_{.0005}$ | $.1235\pm_{.0017}$ | $.2298\pm_{.0031}$ | ○$-.0113\pm_{.0006}$ | $.9950\pm_{.0001}$ | $.9613\pm_{.0005}$ | $.5881\pm_{.1904}$ |
| 4 | LIMEFLDL | ●$.0240\pm_{.0005}$ | ●$.0919\pm_{.0016}$ | ●$.1778\pm_{.0032}$ | ●$.0034\pm_{.0001}$ | ●$.9966\pm_{.0001}$ | ●$.9701\pm_{.0005}$ | ●$.6462\pm_{.1501}$ |
| 5 | PULIME | $.1966\pm_{.0023}$ | $.9352\pm_{.0052}$ | $1.997\pm_{.0130}$ | ○$-.0500\pm_{.0284}$ | $.9129\pm_{.0022}$ | $.7106\pm_{.0031}$ | $.4464\pm_{.0906}$ |
| 5 | LIMEFLDL | ●$.1317\pm_{.0022}$ | ●$.8892\pm_{.0079}$ | ●$1.800\pm_{.0214}$ | ●$.7059\pm_{.0409}$ | ●$.9132\pm_{.0029}$ | ●$.8058\pm_{.0033}$ | ●$.5979\pm_{.1281}$ |
| 6 | PULIME | $.3584\pm_{.0036}$ | $1.995\pm_{.0025}$ | $5.330\pm_{.0110}$ | ○$-.5122\pm_{.0087}$ | $.7999\pm_{.0036}$ | $.1487\pm_{.0039}$ | $.4387\pm_{.0440}$ |
| 6 | LIMEFLDL | ●$.1692\pm_{.0030}$ | ●$1.694\pm_{.0095}$ | ●$4.139\pm_{.0309}$ | ●$1.200\pm_{.0590}$ | ●$.8840\pm_{.0038}$ | ●$.7141\pm_{.0036}$ | ●$.5481\pm_{.0985}$ |
| 7 | PULIME | $.0051\pm_{.0001}$ | ●$.0192\pm_{.0004}$ | ●$.0385\pm_{.0008}$ | $.0003\pm_{.0002}$ | ●$.9998\pm_{.0000}$ | ●$.9935\pm_{.0001}$ | ●$.8661\pm_{.0829}$ |
| 7 | LIMEFLDL | ●$.0049\pm_{.0001}$ | $.0217\pm_{.0010}$ | $.0429\pm_{.0020}$ | ●$.0001\pm_{.0000}$ | $.9998\pm_{.0000}$ | $.9931\pm_{.0002}$ | $.8292\pm_{.0555}$ |
| 8 | PULIME | ●$.0063\pm_{.0001}$ | ●$.0372\pm_{.0011}$ | ●$.0753\pm_{.0015}$ | ●$.0003\pm_{.0001}$ | ●$.9997\pm_{.0000}$ | ●$.9912\pm_{.0001}$ | $.2149\pm_{.1040}$ |
| 8 | LIMEFLDL | $.0081\pm_{.0001}$ | $.0506\pm_{.0008}$ | $.1033\pm_{.0013}$ | $.0006\pm_{.0000}$ | $.9995\pm_{.0000}$ | $.9881\pm_{.0001}$ | ●$.8740\pm_{.0510}$ |

distribution. The participants are shown both the original image and the generated interpretations. They are then asked to assess the degree of deviation in the individual label interpretations and provide reasons for the model's incorrect predictions based on these interpretations, along with their own reasoning. The participants first evaluate whether the reasons are reasonable. If so, they then rate the usefulness of the explanation on a scale from 1 to 5, where 1 meant that the explanation offered no insight into the model's incorrect prediction and 5 meant the explanation is extremely helpful in understanding the incorrect prediction.

### 4.5. Results and discussion

Tables 2 and 3 present the detailed experimental results for the fidelity and consistency metrics of the LDL algorithm. These results are based on six tabular datasets and two image datasets. For each data set, we performed 10-fold cross-validations and recorded the mean and standard deviation for each evaluation metric.

Based on the results of the numerical experiments, the following observations can be made: (1) alpha and cold dataset shows better results both on multiple fidelity metrics and with different black-box modeling algorithms. The rankings of the two interpretation algorithms have their own strengths because the label distributions of these datasets are approximately uniformly distributed. This makes it easy to approximate whether the parallel interpretation or the interpretation distribution. For data sets JAFFE, 3DFE, Emo and Scene, regardless of the LDL algorithm used, LIMEFLDL outperforms PULIME. In addition, the label distributions in

these datasets are derived from subjective labeling or ranking, making them more representative. The results from the two image datasets show better performance compared to the tabular datasets. This is due to the image dataset being characterized by pixel blocks, while the tabular dataset has feature columns with a partial range, leading to higher error in the latter. (2) Our interpretation algorithm generally outperforms PULIME in terms of consistency metric (Jaccard index), which aligns with the stability shown in the theoretical section of this paper. (3) Anomalous data are observed in the KL metric under the PULIME, with its KL divergence being negative. This occurs because the PULIME fits the label distribution individually. Although this approach better captures the description degree of each label, it can result in the sum of the description degree exceeding 1. This leads to the possibility of $\mathbb{P} = \frac{\sum_j^r \Theta_j(\boldsymbol{\xi})-1}{\sum_j^r \Theta_j(\boldsymbol{\xi})}$, which causes the final KL divergence to be negative (see the Appendix B.6 for the overall proof), where $\Theta_j(\boldsymbol{\xi})$ represents the descriptive measure of each label in the fit. Experiments on two additional black-box models are provided in the Appendix A.8.

Figure 3 presents a qualitative evaluation method based on human visual perception to assess the consistency of the interpretation results with human understanding. In Figure 3, purple (a blend of blue and pink) denotes the cases where LIME = LIMEFLDL; blue denotes the cases where LIME > LIMEFLDL; pink denotes the cases where LIMEFLDL > LIME. Participants are shown images with accurate model predictions along with interpretations of PULIME and LIMEFLDL. On average, PULIME receives a score of 2.1, while LIMEFLDL receives a score of 3.1. Thus, LIMEFLDL outperforms PULIME by 1.0 points.

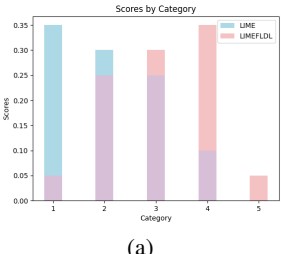
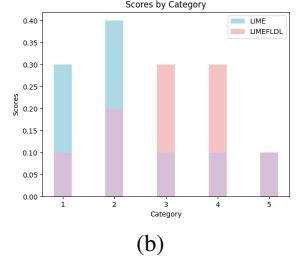

(a)                    (b)

*Figure 3.* Human-interpretability results. Purple (a blend of blue and pink) denotes the cases where LIME = LIMEFLDL; blue denotes the cases where LIME > LIMEFLDL; pink denotes the cases where LIMEFLDL > LIME. (a): Human-interpretability results for accurate predictions. (b): Human-interpretability results for incorrect predictions.

Meanwhile, when participants review images with incorrect model predictions and corresponding interpretations from both methods, PULIME receives an average score of 2.3, while LIMEFLDL scores 3.1. Again, LIMEFLDL achieves a higher score, with a difference of 0.8 points. These results strongly indicate that LIMEFLDL is more effective in explaining model behavior compared to PULIME.

## 5. Conclusions

In this paper, we propose a local interpretable model-agnostic explanation approach for LDL, called LIMEFLDL. This algorithm adapts the original explanation approach to fulfill the constraints of the label distribution learning paradigm. We introduce a new constraint on the form of label distribution and the corresponding feature attribution matrix solution based on the alternating direction method of multipliers. Besides, we give new feature selection functions to improve the interpreting efficiency of model. We then provide a theoretical analysis of the solutions, convergence behaviors, and stability cases of the model. Furthermore, we present a series of properties of our proposed explanation algorithm to facilitate the practical utilization. Finally, we demonstrate the effectiveness of LIMEFLDL by both theoretical analysis and experimental studies.

## Acknowledgements

This research is partially supported by the National Natural Science Foundation of China (62176123, 62476130), and the Natural Science Foundation of Jiangsu Province (BK20242045).

## Impact Statement

This paper makes a significant contribution to the field of interpretable machine learning, which is crucial for the development of LDL. We show that our methods are versatile for any LDL black models (covering at least two data types– tabular data and images). Our research provides the theoretical foundation for further advancements in interpretable LDL. Facilitating the broader adoption of AI technologies in sensitive and impactful domains.

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

# A. More discussions

### A.1. Details of parameter settings

In the fidelity-metric experiments, we set the number of sampled instances $m$ to 100, and select the top 10 ranked features for evaluation on all test sets. For the consistency experiments, to ensure the explanatory model captures sufficient information, we increase $m$ to 1000 and do not impose restrictions on feature rankings. Furthermore, in all experiments, the lagrange multiplier $u^{(0)}$ is initialized as an all-zero vector. The penalty coefficients $\lambda$, $v$ and $\rho$ are set to 0.01.

Parallel Utilization of LIME (PULIME) decomposes label distribution into a series of single-label. For each label, the explanation task is performed independently. Specifically, the sampling size $m$ and weighting parameters align with LIMEFLDL. Other hyperparameters retain its default settings.

**Feature transformation.** For image data, the first step is to crop each image to a size of (256, 256, 3). The image is then segmented into superpixels using the Quickshift method from scikit-image, with the following parameters: kernel-size=4, max-dist=200, ratio=0.2, and random-seed=2024. These settings align with the default configuration in LIME, except for the modified random seed. This ensures that the same images produce the same superpixels, reducing computational instability in the interpretation process. However, different images are still segmented in different ways. For tabular data, the parameters are set as follows: discretizer="entropy", random-seed=2024, and discretize-continuous='True'. The decision tree method is used to adaptively bin feature columns. Fixed random seeds ensure consistency while guaranteeing that each feature column is involved in the binning process.

**Computing Explanations.** For image explanations, the masking operation follows the original LIME settings. The hide-color parameter is set to None, meaning that the average value of each superpixel serves as its reference when that superpixel is removed. For distance weights, cosine is used for image datasets, while the $L_2$ norm is applied for tabular datasets. To balance accuracy and efficiency, the maximum number of iterations for L-BFGS is set to 50. In addition, all random seed parameters are fixed to ensure reproducibility and consistency across runs.

### A.2. Supplementary notes on human experimentation

We conducted the human evaluation with full adherence to ethical standards. Here is the required information: Our design mitigated bias via 10 postgraduate students participated in the study, with 7 specializing in machine learning and the remaining 3 focusing on other areas within computer science. The evaluation was conducted through in-person scoring sessions. Each participant independently evaluated 20 randomly selected cases from our test set. All participants signed informed consent forms outlining data usage. No personally identifiable information was collected. Raw responses were aggregated to prevent individual identification. Random recruitment (ML/non-ML backgrounds), independent evaluation of 20 randomized interpretation results, and high inter-rater agreement despite participant diversity.

### A.3. Sampling weight

For image data, a sampled instance $z_i$, can differ significantly from the original image. For instance, if most of the elements of $z_{iq}$ are zero, $z_i$ will be close to the mean. When constructing a local model, each sampled instance differs slightly from the instance to be interpreted. Therefore, we treat the proximity as a weight $\pi(z_i)$,

$$\forall 1 \leq i \leq m, \quad \pi(z_i) = \exp\left(\frac{-d_{\cos}(1, z_i)^2}{\sigma^2}\right), \tag{16}$$

where $\sigma > 0$ is the kernel width parameter and $d_{\cos}$ is the cosine distance.

$$\forall u, v \in \mathbb{R}^d, \quad d_{\cos}(u, v) = 1 - \frac{u^\top v}{\|u\| \cdot \|v\|}. \tag{17}$$

We can find that the weights $\pi(z_i)$ depend solely on the number of hyperpixel blocks, each taking the mean of its corresponding channel. For example, if $z_i$ has exactly $s$ elements equal to zero, we have $z_i^\top 1 = m - s$, and $\|z_i\| = \sqrt{m - s}$. So $\|1\| = \sqrt{m}$, according to Equation (16), we deduce that $\pi(z_i) = \rho(\frac{s}{m})$,

$$\forall a \in [0, 1], \quad \rho(a) = \exp\left(\frac{-(1 - \sqrt{1 - a})^2}{\sigma^2}\right). \tag{18}$$

---

**Algorithm 2** Get Summary Statistics: Getting summary statistics from training data

---

**Input**: Train set $\mathcal{X} = \{\zeta_1, \ldots, \zeta_n\}$, number of bins $p$.

1: **for** $j = 1$ to $f$ **do**
2:     **for** $b = 0$ to $p$ **do**
3:        $q_{jb} \leftarrow \text{Quantile}\left(\zeta_{1j}, \ldots, \zeta_{fj}; b/p\right)$
4:     **end for**
5:     **for** $b = 1$ to $p$ **do**
6:        $\mathcal{S} \leftarrow \{\zeta_{ij} \text{ s.t. } q_{jb-1} < \zeta_{ij} \le q_{jb}\}$
7:        $\mu_{jb} \leftarrow \text{Mean}(\mathcal{S})$
8:        $\sigma_{jb}^2 \leftarrow \text{Var}(\mathcal{S})$
9:     **end for**
10: **end for**
11: **return** $q_{jb}, \mu_{jb}$, and $\sigma_{jb}^2$ for $1 \le j \le f$ and $1 \le b \le p$

---

Tabular data sampling begins by dividing the input space of each column of features in the training set into bins. These bins are used to create interpretable features. The method not only uses the ranges of the features, but also computes the mean and variance of the training set data once transformed into interpretable features. These values are then used to generate subsequent sampling instances. For detailed specifics, please refer to Algorithm 2.

**Boundary extraction.** The boundaries are determined by taking the quantiles at level $\frac{b}{p}$ for $b \in \{0, 1, \ldots, p\}$ . In other words, the $q_{jb}$ are such that

$$\forall 1 \le j \le f, \forall 1 \le b \le p, \quad \frac{1}{m} \sum_{i=1}^{m} \mathbf{1}_{\zeta_{ij} \in [q_{jb-1}, q_{jb}]} \approx \frac{1}{p}, \tag{19}$$

where, for each $1 \le j \le f$, $q_{j0} = \min_{1 \le i \le m} \zeta_{i,j}$, and $q_{jp} = \max_{1 \le i \le m} \zeta_{i,j}$. On the other hand, the indices of the bins are used to sample new samples $z_1, \ldots, z_m$. For $1 \le i \le m$, each new sample $z_i$ is sampled dimension-wise independently: $z_{jq}$ follows the distribution of a truncated Gaussian random variable with parameters $q_{jb_{ij}-1}, q_{jb_{ij}}, \mu_{jb_{ij}}$ and $\sigma_{jb_{ij}}^2$. So, we condition, when falling into the $b$-th bin, the probability density function of $z_{jq}$ is given by the following expression.

$$\rho_{jb}(t) = \frac{1}{\sigma_{jb}\sqrt{2\pi}} \cdot \frac{\exp\left(\frac{-(t-\mu_{jb})^2}{2\sigma_{j,b}^2}\right)}{\Phi\left(r_{jb}\right) - \Phi\left(\ell_{jb}\right)} \mathbf{1}_{t \in [q_{jb-1}, q_{jb}]}, \tag{20}$$

where we let $\ell_{jb} = \frac{q_{jb-1}-\mu_{jb}}{\sigma_{jb}}$ and $r_{jb} = \frac{q_{jb}-\mu_{jb}}{\sigma_{jb}}$ and $\Phi$ is the cumulative distribution function of a standard Gaussian random variable.

In the tabular data, the weights of the newly sampled instances are given by:

$$\pi(z_i) = \exp\left(-\frac{\|\mathbf{1} - z_i\|^2}{\sigma^2}\right) = \exp\left(-\frac{1}{\sigma^2} \sum_{j=1}^{f} \mathbf{1}_{b_{ij} \ne b_j}\right), \tag{21}$$

where $\|\cdot\|$ denotes the $L_2$ norm, and $\sigma$ is the kernel width parameter. This weight calculation is based on how many coordinates the bins of the sampled instance differ from those of the instance to be explained, followed by the application of exponential scaling. If all bins are identical, the weight is 1. On the other hand, if the sampled instance is far from the instance to be explained, $\pi(z_i)$ can be very small. Here, far means that $Z$ does not belong to the same bins as the instance to be explained, and the bandwidth parameter $\nu$ is fixed at $\sqrt{0.75f}$.

**Limitations.** If two interpretations instances $A_1$ and $A_2$ are very close, they may share the same bin indices and, consequently, the same $A$. On the other hand, if $A_1$ and $A_2$ are close but do not share the same bin indices, their explanations may differ significantly. From a practical perspective, if one believes that the exact values of $A$ are important, increasing bins is a solution. This results in thinner bins per dimension. In the limit, setting $p = 0$, no discretization is also an option.

## A.4. Convergence of LIMEFLDL

Figure 4 extends the experimental analysis in the main text by incorporating two additional common LDL black-box models: AA-KNN and MEM. In these experiments, all data sets from the main text are used. The number of sampling instances increases sequentially from 100, 500, 1000, to 5000, with the top-20 average Jaccard index serving as the evaluation metric. Notably, the results in Figure 4 align closely with those in Figure 1(a). As the number of sampling instances increases, LIMEFLDL consistently converges, regardless of the dataset or the choice of black-box model.

## A.5. Stability of LIMEFLDL

To introduce variability in the black-box model's predictions while avoiding overly biased distribution, we adjusted the number of iteration rounds. For RBF-LDL-LRR and LDL-LRR, the iteration rounds were varied from 10 to 500. For the AA-KNN model, the number of clusters was modified from 1 to 5. Additionally, we calculated the average difference in the descriptiveness of each label between each prediction and the best prediction. The top-20 average Jaccard index was used as the evaluation metric. As shown in Figure 5, the results align closely with those in Figure 1(b). As the average difference in label descriptiveness between each model's predictions and the best model's predictions decreases, the explanations generated by the models become increasingly consistent with those of the best model. These findings demonstrate that smaller perturbations in the black-box model lead to more consistent interpretations across different models.

## A.6. Attribution of additive features in LIMEFLDL

To ensure the additive feature attribution holds, we add features one at a time. For image data, we first mask each superpixel using the mean of the three color channels. Secondly, based on the interpretation results of the original image, we compute the feature attribution distribution for each superpixel block. We take the absolute values of these attributions and sum them. A larger value indicates a greater impact on the overall label distribution. Thirdly, we select the top 10 superpixel blocks and incrementally add the highest-ranked block from the original image, starting with a mean mask for each block. The masking operation flow for this process is shown in Figure 11. Fourthly, we observe the distance between the local prediction and the original image's local prediction, which is represented using fidelity. For the tabular dataset, we compute the top 10 features of the instances to be interpreted and add the features one at a time. For unselected features, we set the corresponding column to 0, indicating it was not chosen for the bin. The results in Figure 7, show that, as features are gradually added, the individual fidelity metrics increase, meaning the predictions get closer to the original instance. This indicates that LIMEFLDL's additive feature attribution is correct.

## A.7. Dummy features of LIMEFLDL

Due to the harsh conditions of the virtual features, it is difficult to find feature attribution distribution that do not affect the whole label distribution in any way. However, it is possible to find the feature with the least impact based on the feature attribution ranking. Therefore, we choose to focus on the feature with the lowest ranks. For the image data, we first identify the lowest-ranked superpixel block. To demonstrate that the feature is almost false, we apply a complex process: we mask the superpixel, and add a green border. A specific example is shown in Figure 6. We then evaluate the fidelity and consistency of both the original and processed images. Consistency results are in Figure 8 and Figure 9, and fidelity result is in Figure 10.

## A.8. Comparison of fidelity and consistency metrics in AA-KNN and MEM

Our goal is to enable our explanation algorithm to interpret complex black-box models. However, if the explanations for simpler models do not provide sufficient fidelity, it is challenging to trust explanations for more complex black-box models. Therefore, in addition to comparing the fidelity and consistency rankings of our algorithm with PULIME on RBF-LDL-LRR and LDL-SCL black-box models, we also conducted experiments on simpler black-box models such as AA-KNN and Maximum Entropy. These models have relatively simpler processes. AA-KNN integrates a clustering algorithm within the model, while MEM employs KL divergence as its loss function. Experimental results show that our algorithm performs better on simpler black-box models than on more complex ones. Furthermore, in most datasets, our method outperforms PULIME in both fidelity and consistency metrics.

## A.9. The ranking information in the comparison results between LIMEFLDL and PULIME

We give ranking statistics to illustrate the extent to which LIMEFLDL is ahead of PULIME in terms of evaluation metrics, the results are in Table 4, from the result, we can see that LIMEFLDL leads the rankings for the vast majority of measures.

*Table 4.* The ranking information in the comparison results between LIMEFLDL and PULIME.

| Model | Algorithm | Chebyshev | Clark | Canberra | KL | Cosine | Intersection | Jaccard |
|---|---|---|---|---|---|---|---|---|
| RBF-LDL-LRR | LIME | 1.75 | 1.5 | 1.625 | 2 | 1.625 | 1.625 | 1.875 |
| | LIMEFLDL | 1.25 | 1.5 | 1.375 | 1 | 1.375 | 1.375 | 1.125 |
| LDL-SCL | LIME | 1.625 | 1.5 | 1.5 | 1.875 | 1.5 | 1.5 | 1.625 |
| | LIMEFLDL | 1.375 | 1.5 | 1.5 | 1.125 | 1.5 | 1.5 | 1.375 |
| AA-KNN | LIME | 1.625 | 1.625 | 1.625 | 1.5 | 1.625 | 1.625 | 1.75 |
| | LIMEFLDL | 1.375 | 1.375 | 1.375 | 1.5 | 1.375 | 1.375 | 1.25 |
| MEM | LIME | 1.75 | 1.75 | 1.75 | 1.875 | 1.75 | 1.75 | 1.75 |
| | LIMEFLDL | 1.25 | 1.25 | 1.25 | 1.125 | 1.125 | 1.25 | 1.25 |

## A.10. The computational complexity of the LIMEFLDL algorithm compared to PULIME

We analyze the computational complexity of LIMEFLDL and PULIME. For LIMEFLDL, sampling and forming weight matrix $\Pi$ requires $O(m)$. Matrix multiplication contributes $O(mfr)$, black-box model inference accounts for $O(mrk)$. Element-wise matrix subtraction and F-paradigm operations each add $O(mr)$, with regularization terms contributing $O(fr)$. The total complexity is $O(mfr + mrk + mr + mr + fr + m)$. PULIME's per-label workflow involves sampling ($O(m)$), vector operations ($O(m)$ for subtraction/squaring and $O(mf)$ for multiplication), black-box inference ($O(mrk)$), regularization ($O(f)$). With parallel computation, its total complexity is $O(mrf + mkr^2 + mr + mr + mr + rf)$. PULIME's ridge regression solver has $O(f^3 + mf^2)$ complexity, LIMEFLDL uses L-BFGS optimization with $O(T(zr + mf))$ cost, where $z \approx 5$–20 (stored gradient pairs) and $T$ denotes iterations.

## A.11. R²-score aligned with LIME metric and expanded baselines

We conducted supplementary experiments using R²-score aligned with LIME metric and expanded baselines to GLIME and DLIME, all evaluated under default parameters. The result is in Table 5. It can be seen that LIMEFLDL leads in the original fidelity metrics for most of the table and image datasets.

*Table 5.* R²-score aligned with LIME metric and expanded baselines.

| Model | Index | LIME | DLIME | GLIME | LIMEFLDL |
|---|---|---|---|---|---|
| LDL-SCL | 1 | 0.16 | 0.15 | 0.16 | 0.67 |
| | 2 | 0.15 | 0.15 | 0.16 | 0.67 |
| | 3 | 0.48 | 0.35 | 0.34 | 0.58 |
| | 4 | 0.39 | 0.33 | 0.34 | 0.40 |
| | 5 | 0.34 | 0.25 | 0.25 | 0.30 |
| | 6 | 0.37 | 0.34 | 0.32 | 0.38 |
| | 7 | 0.94 | - | 0.95 | 0.99 |
| | 8 | 0.89 | - | 0.89 | 0.98 |
| RBF-LDL-LRR | 1 | 0.17 | 0.17 | 0.16 | 0.99 |
| | 2 | 0.16 | 0.15 | 0.17 | 0.99 |
| | 3 | 0.34 | 0.33 | 0.33 | 0.36 |
| | 4 | 0.25 | 0.31 | 0.29 | 0.36 |
| | 5 | 0.26 | 0.47 | 0.36 | 1.00 |
| | 6 | 0.52 | 0.49 | 0.47 | 0.99 |
| | 7 | 0.93 | - | 0.96 | 0.99 |
| | 8 | 0.67 | - | 0.94 | 0.99 |

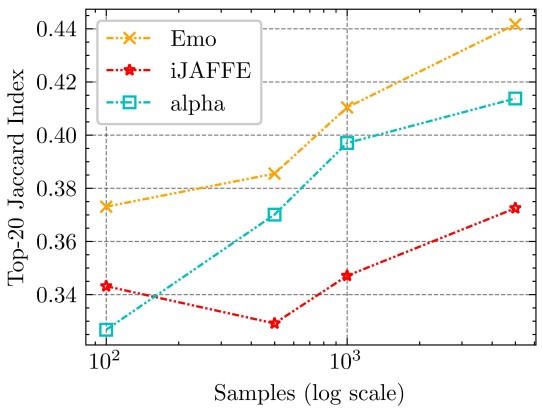
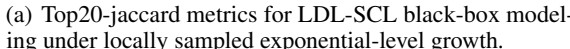

(a) Top20-jaccard metrics for LDL-SCL black-box modeling under locally sampled exponential-level growth.

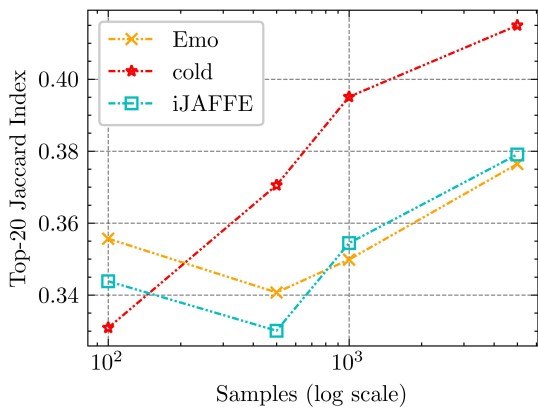

(b) Top20-jaccard metrics for AA-KNN black-box modeling under locally sampled exponential-level growth.

*Figure 4.* Visualization the convergence of explanation algorithms.

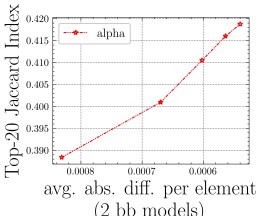

(a) Top-20 jaccard metrics on the Yeast_alpha tabular dataset.

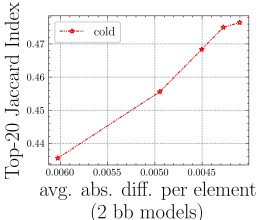

(b) Top-20 jaccard metrics on the Yeast_cold tabular dataset.

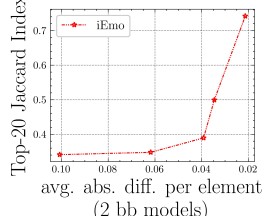

(c) Top-20 jaccard metrics on the Emotion6 image dataset.

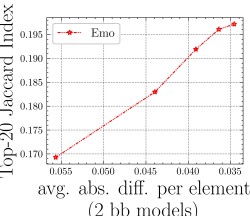

(d) Top-20 jaccard metrics on the Emotion6 tabular dataset.

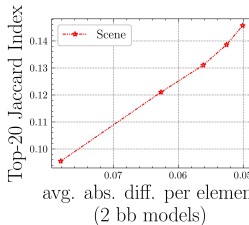

(e) Top-20 jaccard metrics on the Natural Scene tabular dataset.

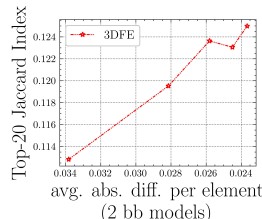

(f) Top-20 jaccard metrics on the SBU_3DFE tabular dataset.

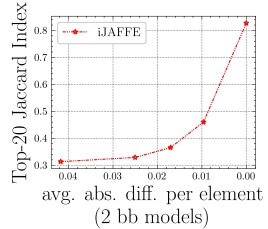

(g) Top-20 jaccard metrics on the JAFFE image dataset.

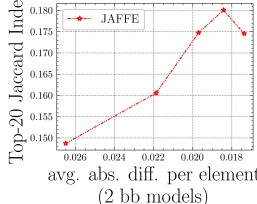

(h) Top-20 jaccard metrics on the JAFFE tabular dataset.

*Figure 5.* Stability evaluation with AA-KNN black-box models.

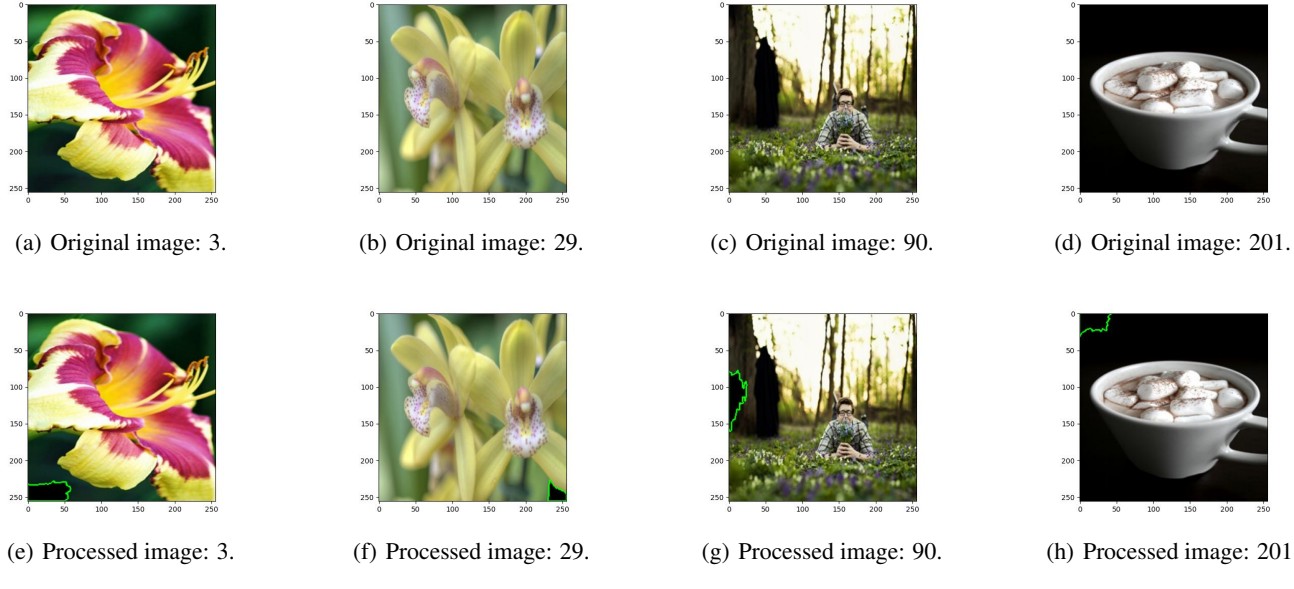

(a) Original image: 3.  (b) Original image: 29.  (c) Original image: 90.  (d) Original image: 201.

(e) Processed image: 3.  (f) Processed image: 29.  (g) Processed image: 90.  (h) Processed image: 201.

*Figure 6.* Dummy feature processing in image data.

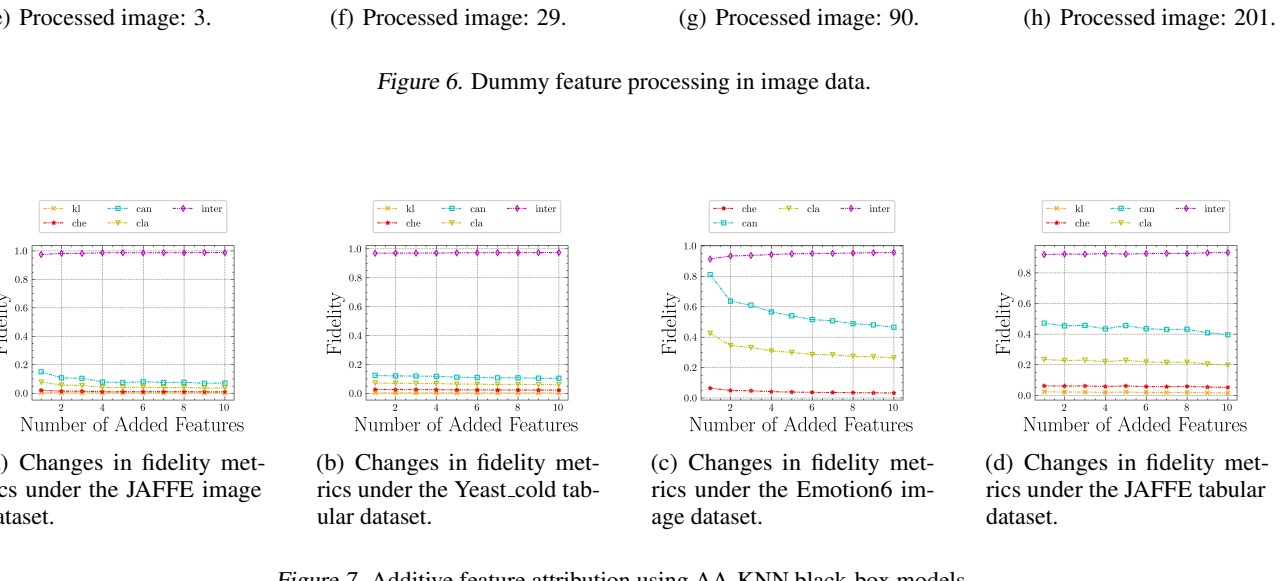

(a) Changes in fidelity metrics under the JAFFE image dataset.

(b) Changes in fidelity metrics under the Yeast_cold tabular dataset.

(c) Changes in fidelity metrics under the Emotion6 image dataset.

(d) Changes in fidelity metrics under the JAFFE tabular dataset.

*Figure 7.* Additive feature attribution using AA-KNN black-box models.

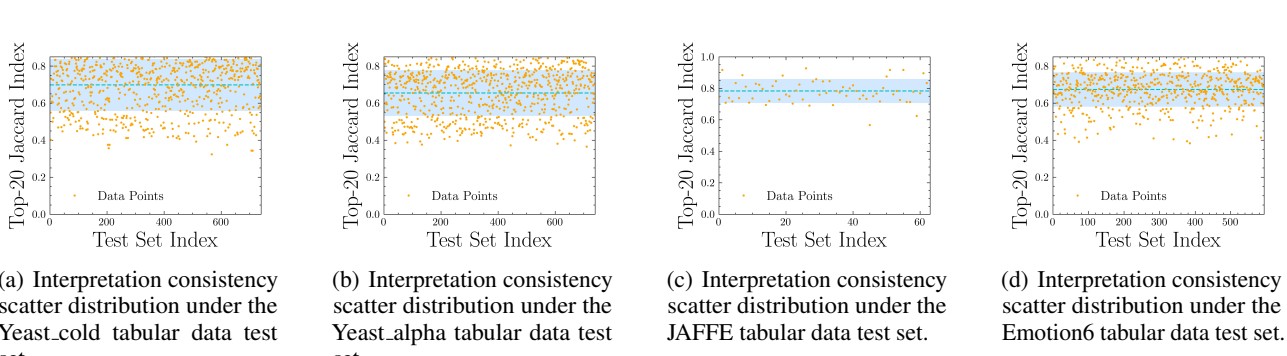

(a) Interpretation consistency scatter distribution under the Yeast_cold tabular data test set.

(b) Interpretation consistency scatter distribution under the Yeast_alpha tabular data test set.

(c) Interpretation consistency scatter distribution under the JAFFE tabular data test set.

(d) Interpretation consistency scatter distribution under the Emotion6 tabular data test set.

*Figure 8.* Dummy feature attribution of the LDL-SCL black-box model.

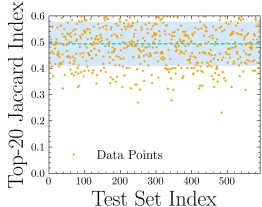

(a) Interpretation consistency scatter distribution under the Yeast_cold tabular data test set.

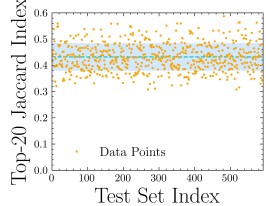

(b) Interpretation consistency scatter distribution under the Yeast_alpha tabular data test set.

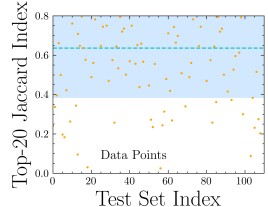

(c) Interpretation consistency scatter distribution under the JAFFE image data test set.

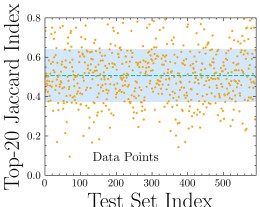

(d) Interpretation consistency scatter distribution under the Emotion6 tabular data test set.

*Figure 9.* Dummy feature attribution of the AA-KNN black-box model.

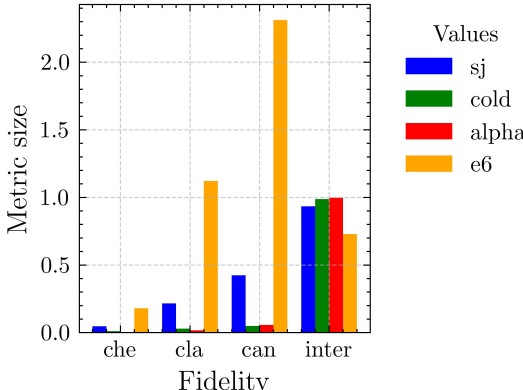

(a) Modifying the lowest-ranked feature using the LDL-SCL black-box model.

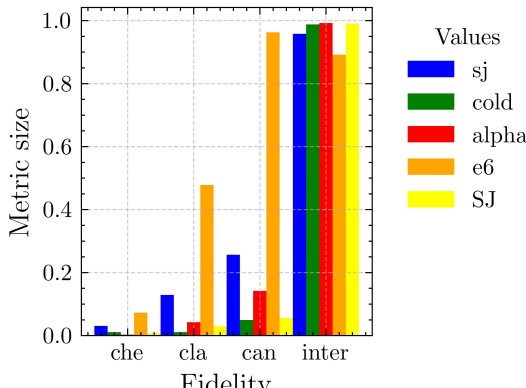

(b) Modifying the lowest-ranked feature using the AA-KNN black-box model.

*Figure 10.* Dummy feature attribution under fidelity metrics.

# B. Proofs

## B.1. The range of $\left\|B^{-1}\right\|_F$, $\left\|C\right\|_F$, and the specific value of $\left\|E\right\|_F$

First, the specific form of the $B$ matrix is presented,

$$B = \begin{bmatrix} \frac{1}{m}\sum_i \pi\left(\boldsymbol{z_i}\right)\left(z_{i1}\right)^2 & \frac{1}{m}\sum_i \pi\left(\boldsymbol{z_i}\right)z_{i1}z_{i2} & \cdots & \frac{1}{m}\sum_i \pi\left(\boldsymbol{z_i}\right)z_{i,1}z_{if+1} \\ \frac{1}{m}\sum_i \pi\left(\boldsymbol{z_i}\right)z_{i1}z_{i2} & \frac{1}{m}\sum_i \pi\left(\boldsymbol{z_i}\right)\left(z_{i2}\right)^2 & \cdots & \frac{1}{m}\sum_i \pi\left(\boldsymbol{z_i}\right)z_{i2}z_{if+1} \\ \vdots & \vdots & \ddots & \vdots \\ \frac{1}{m}\sum_i \pi\left(\boldsymbol{z_i}\right)z_{i1}z_{if+1} & \frac{1}{m}\sum_i \pi\left(\boldsymbol{z_i}\right)z_{i2}z_{if+1} & \cdots & \frac{1}{m}\sum_i \pi\left(\boldsymbol{z_i}\right)\left(z_{if+1}\right)^2 \end{bmatrix}. \tag{22}$$

*Table 6.* The effectiveness of two interpretation algorithms is evaluated on different datasets, using fidelity and consistency metrics under black-box modeling with AA-KNN. Results are marked with ● for better outcomes and ○ for anomalous data.

| Index | Algorithms | Chebyshev ↓ | Clark ↓ | Canberra ↓ | KL ↓ | Cosine ↑ | Intersection ↑ | Jaccard ↑ |
|---|---|---|---|---|---|---|---|---|
| 1 | PULIME | ● .0025±.0000 | ● .0400±.0003 | ●.1319±.0008 | ●.0001±.0001 | ●.9998±.0000 | ●.9927±.0000 | .4206±.0462 |
| 1 | LIMEFLDL | .0027±.0000 | .0447±.0002 | .1485±.0008 | .0002±.0000 | .9998±.0000 | .9918±.0000 | ●.4743±.0816 |
| 2 | PULIME | ●.0105±.0001 | ● .0278±.0003 | ● .0477±.0006 | ●.0003±.0001 | ● .9996±.0000 | ●.9881±.0002 | .4908±.0675 |
| 2 | LIMEFLDL | .0107±.0001 | .0291±.0002 | .0504±.0004 | .0004±.0000 | .9995±.0000 | .9875±.0001 | ●.5183±.1561 |
| 3 | PULIME | .0350±.0017 | .1466±.0056 | .2841±.0095 | ○ -.0306±.0046 | .9940±.0005 | .9533±.0016 | .4261±.1107 |
| 3 | LIMEFLDL | ● .0242±.0011 | ● .0996±.0052 | ●.1977±.0104 | ● .0038±.0004 | ●.9964±.0003 | ●.9675±.0017 | ●.4322±.0821 |
| 4 | PULIME | .0576±.0007 | .2080±.0011 | .3904±.0026 | ○ -.0662±.0014 | .9886±.0001 | .9327±.0005 | ●.4167±.0555 |
| 4 | LIMEFLDL | ● .0322±.0005 | ● .1256±.0019 | ● .2493±.0037 | ● .0062±.0001 | ●.9939±.0001 | ● .9581±.0006 | .3845±.0602 |
| 5 | PULIME | .0794±.0012 | .6438±.0080 | 1.225±.0153 | ○ -.0054±.0040 | .9723±.0009 | .8835±.0017 | .3742±.0606 |
| 5 | LIMEFLDL | ●.0554±.0011 | ●.5206±.0087 | ●.9571±.0160 | ● .0389±.0037 | ●.9839±.0004 | ● .9173±.0013 | ●.4229±.0800 |
| 6 | PULIME | .1505±.0020 | 1.686±.0033 | 3.777±.0121 | ●.0088±.0126 | .9320±.0012 | .7791±.0020 | .0650±.0818 |
| 6 | LIMEFLDL | ● .1094±.0015 | ● 1.610±.0061 | ●3.573±.0142 | .1494±.0053 | ●.9523±.0011 | ●.8408±.0018 | ●.1565±.0203 |
| 7 | PULIME | .0060±.0007 | .0249±.0027 | .0467±.0043 | .0006±.0003 | .9996±.0001 | .9924±.0007 | ●.3553±.1402 |
| 7 | LIMEFLDL | ● .0049±.0002 | ● .0189±.0011 | ●.0366±.0023 | ● .0003±.0000 | ● .9997±.0000 | ●.9937±.0003 | .3516±.0750 |
| 8 | PULIME | ●.0201±.0003 | ●.1626±.0023 | ●.2816±.0025 | ● .0065±.0009 | ●.9966±.0001 | ●.9735±.0003 | .1833±.1249 |
| 8 | LIMEFLDL | .0215±.0003 | .2030±.0034 | .3423±.0053 | .0083±.0019 | .9958±.0001 | .9707±.0004 | ●.2844±.0929 |

The expected value of each element in the matrix $B$ is:

$$
\begin{aligned}
\mathbb{E}\left[\pi\left(\boldsymbol{z_i}\right)\left(z_{ij}\right)^2\right] &= \sum_{k=0}^{f} e^{\frac{(k-f)}{\sigma^2}} \frac{k}{f} \frac{f!}{2^f(f-k)!k!} \\
&= \sum_{k=0}^{f} e^{\frac{(k-f)}{\sigma^2}} \frac{(f-1)!}{2^f(f-k)!(k-1)!} \\
&= \sum_{k=0}^{f} e^{\frac{(k-1)}{\sigma^2}} e^{\frac{(1-f)}{\sigma^2}} \frac{(f-1)!}{2^f(f-k)!(k-1)!} \\
&= e^{(1-f)/\sigma^2} \frac{\left(1+e^{\frac{1}{\sigma^2}}\right)^{f-1}}{2^f} \\
&= \frac{\left(1+e^{-\frac{1}{\sigma^2}}\right)^{f-1}}{2^f},
\end{aligned}
\tag{23}
$$

similarly,

$$
\begin{aligned}
\mathbb{E}[\pi(\boldsymbol{z_i})(z_{ij}z_{it})] &= \sum_{k=0}^{f} e^{\frac{(k-f)}{\sigma^2}} \frac{k(k-1)}{f(f-1)} \frac{f!}{2^f(f-k)!k!} \\
&= \sum_{k=0}^{f} e^{\frac{(k-f)}{\sigma^2}} \frac{(f-2)!}{2^f(f-k)!(k-2)!} \\
&= \sum_{k=0}^{f} e^{\frac{(k-2)}{\sigma^2}} e^{\frac{(2-f)}{\sigma^2}} \frac{(f-2)!}{2^f(f-k)!(k-2)!} \\
&= e^{(2-f)/\sigma^2} \frac{\left(1+e^{\frac{1}{\sigma^2}}\right)^{f-2}}{2^f} \\
&= \frac{\left(1+e^{-\frac{1}{\sigma^2}}\right)^{f-2}}{2^f}.
\end{aligned}
\tag{24}
$$

*Table 7.* The effectiveness of two interpretation algorithms is evaluated on different datasets, using fidelity and consistency metrics under black-box modeling with MEM. Results are marked with ● for better outcomes and ○ for anomalous data.

| Index | Algorithms | Chebyshev ↓ | Clark ↓ | Canberra ↓ | KL ↓ | Cosine ↑ | Intersection ↑ | Jaccard ↑ |
|---|---|---|---|---|---|---|---|---|
| 1 | PULIME | ●$.0008 \pm_{.0000}$ | ●$.0127 \pm_{.0001}$ | ●$.0417 \pm_{.0004}$ | ○$-.0001 \pm_{.0000}$ | ●$.9999 \pm_{.0000}$ | ●$.9976 \pm_{.0000}$ | $.6612 \pm_{.1533}$ |
| 1 | LIMEFLDL | $.0009 \pm_{.0000}$ | $.0157 \pm_{.0002}$ | $.0523 \pm_{.0006}$ | ●$.0000 \pm_{.0000}$ | $.9999 \pm_{.0000}$ | $.9971 \pm_{.0000}$ | ●$.7029 \pm_{.1506}$ |
| 2 | PULIME | ●$.0034 \pm_{.0001}$ | ●$.0087 \pm_{.0001}$ | ●$.0149 \pm_{.0003}$ | ○$-.0001 \pm_{.0001}$ | ●$.9999 \pm_{.0000}$ | ●$.9962 \pm_{.0001}$ | $.7324 \pm_{.1984}$ |
| 2 | LIMEFLDL | $.0036 \pm_{.0001}$ | $.0097 \pm_{.0001}$ | $.0168 \pm_{.0002}$ | ●$.0001 \pm_{.0000}$ | $.9999 \pm_{.0000}$ | $.9958 \pm_{.0001}$ | ●$.7883 \pm_{.1601}$ |
| 3 | PULIME | $.0721 \pm_{.0033}$ | $.3030 \pm_{.0111}$ | $.5723 \pm_{.0168}$ | $.0612 \pm_{.0149}$ | $.9810 \pm_{.0015}$ | $.9084 \pm_{.0031}$ | $.5218 \pm_{.0732}$ |
| 3 | LIMEFLDL | ●$.0470 \pm_{.0032}$ | ●$.2198 \pm_{.0130}$ | ●$.4296 \pm_{.0241}$ | ●$.0205 \pm_{.0146}$ | ●$.9874 \pm_{.0016}$ | ●$.9349 \pm_{.0037}$ | ●$.6763 \pm_{.0884}$ |
| 4 | PULIME | $.1158 \pm_{.0020}$ | $.3675 \pm_{.0034}$ | $.6878 \pm_{.0059}$ | ○$-.1023 \pm_{.0084}$ | $.9651 \pm_{.0005}$ | $.8712 \pm_{.0013}$ | $.5213 \pm_{.1396}$ |
| 4 | LIMEFLDL | ●$.0544 \pm_{.0010}$ | ●$.2096 \pm_{.0035}$ | ●$.4074 \pm_{.0072}$ | ●$.0188 \pm_{.0022}$ | ●$.9828 \pm_{.0006}$ | ●$.9323 \pm_{.0013}$ | ●$.6152 \pm_{.1426}$ |
| 5 | PULIME | $.1529 \pm_{.0026}$ | $.8101 \pm_{.0103}$ | $1.703 \pm_{.0206}$ | ○$-.0784 \pm_{.0160}$ | $.9344 \pm_{.0017}$ | $.7758 \pm_{.0026}$ | $.5412 \pm_{.0922}$ |
| 5 | LIMEFLDL | ●$.1051 \pm_{.0017}$ | ●$.7260 \pm_{.0057}$ | ●$1.468 \pm_{.0121}$ | ●$.3255 \pm_{.0289}$ | ●$.9420 \pm_{.0018}$ | ●$.8442 \pm_{.0019}$ | ●$.6344 \pm_{.1243}$ |
| 6 | PULIME | $.4383 \pm_{.0033}$ | $2.113 \pm_{.0034}$ | $5.783 \pm_{.0139}$ | ○$-.6579 \pm_{.0160}$ | $.7737 \pm_{.0023}$ | ○$-.1340 \pm_{.0083}$ | ●$.0699 \pm_{.0823}$ |
| 6 | LIMEFLDL | ●$.2022 \pm_{.0027}$ | ●$1.868 \pm_{.0134}$ | ●$4.692 \pm_{.0451}$ | ●$1.855 \pm_{.0742}$ | ●$.8339 \pm_{.0043}$ | ●$.6454 \pm_{.0042}$ | $.0200 \pm_{.0063}$ |
| 7 | PULIME | $.0086 \pm_{.0004}$ | $.0358 \pm_{.0012}$ | $.0715 \pm_{.0025}$ | $.0005 \pm_{.0000}$ | $.9995 \pm_{.0000}$ | $.9883 \pm_{.0004}$ | $.8755 \pm_{.0639}$ |
| 7 | LIMEFLDL | ●$.0050 \pm_{.0001}$ | ●$.0192 \pm_{.0003}$ | ●$.0384 \pm_{.0007}$ | ●$.0002 \pm_{.0002}$ | ●$.9999 \pm_{.0000}$ | ●$.9936 \pm_{.0001}$ | ●$.9000 \pm_{.1349}$ |
| 8 | PULIME | $.0079 \pm_{.0001}$ | $.0485 \pm_{.0007}$ | $.1008 \pm_{.0013}$ | ●$.0006 \pm_{.0000}$ | $.9995 \pm_{.0000}$ | $.9882 \pm_{.0001}$ | ●$.8580 \pm_{.0580}$ |
| 8 | LIMEFLDL | ●$.0074 \pm_{.0001}$ | ●$.0400 \pm_{.0004}$ | ●$.0828 \pm_{.0009}$ | $.0007 \pm_{.0001}$ | ●$.9997 \pm_{.0000}$ | ●$.9900 \pm_{.0001}$ | $.7670 \pm_{.1061}$ |

Let,

$$\mu_1 = \frac{\left(1 + e^{-\frac{1}{\sigma^2}}\right)^{f-1}}{2^f}, \mu_2 = \frac{\left(1 + e^{-\frac{1}{\sigma^2}}\right)^{f-2}}{2^f}, \tag{25}$$

ignoring the added bias term, the expected value of the $\boldsymbol{B}$ matrix can be expressed as:

$$\boldsymbol{B} = (\mu_1 - \mu_2)\, \boldsymbol{I}_{f \times 1} + \mu_2\left(\mathbf{1}_{f \times 1} \times \mathbf{1}_{1 \times f}\right). \tag{26}$$

Then, the inverse of matrix $\boldsymbol{B}$ can be expressed as:

$$\boldsymbol{B}^{-1} = \left((\mu_1 - \mu_2)\, \boldsymbol{I}_{f \times 1} + \mu_2\left(\mathbf{1}_{f \times 1} \times \mathbf{1}_{1 \times f}\right)\right)^{-1}, \tag{27}$$

at the same time, we can obtain:

$$\begin{aligned}
\frac{1}{2^f} &\leq \frac{\left(1 + e^{-\frac{1}{\sigma^2}}\right)^{f-1}}{2^f} \leq \frac{2^{f-1}}{2^f} = \frac{1}{2}, \\
\frac{1}{2^f} &\leq \frac{\left(1 + e^{-\frac{1}{\sigma^2}}\right)^{f-2}}{2^f} \leq \frac{2^{f-2}}{2^f} = \frac{1}{4}.
\end{aligned} \tag{28}$$

By applying the Sherman-Morrison formula, we obtain:

$$\begin{aligned}
\boldsymbol{B}^{-1} &= \left((\mu_1 - \mu_2)\, \boldsymbol{I}_{f \times f} + \mu_2(\mathbf{1}_{f \times 1} \times \mathbf{1}_{1 \times f})\right)^{-1} \\
&= \frac{1}{\mu_1 - \mu_2}\left(\boldsymbol{I}_{f \times f} + \frac{\mu_2}{\mu_1 - \mu_2}\mathbf{1}_{f \times 1}\mathbf{1}_{1 \times f}^\top\right)^{-1} \\
&= \frac{1}{\mu_1 - \mu_2}\left(\boldsymbol{I}_{f \times f} - \frac{\frac{\mu_2}{\mu_1 - \mu_2}\mathbf{1}_{f \times 1}\mathbf{1}_{1 \times f}^\top}{1 + \frac{\mu_2}{\mu_1 - \mu_2}f}\right).
\end{aligned} \tag{29}$$

Let,

$$\nu_1 = \frac{\mu_1 + (f-2)\mu_2}{(\mu_1 - \mu_2)(\mu_1 + (f-1)\mu_2)}, \quad \nu_2 = -\frac{\mu_2}{(\mu_1 - \mu_2)(\mu_1 + (f-1)\mu_2)}, \tag{30}$$

then, $\boldsymbol{B}^{-1} = (\nu_1 - \nu_2)\, \boldsymbol{I}_{f \times f} + \nu_2\left(\mathbf{1}_{f \times 1} \times \mathbf{1}_{1 \times f}\right)$ in that case, we get:

$$\left\|\boldsymbol{B}^{-1}\right\|_F^2 = f\nu_1^2 + \left(f^2 - f\right)\nu_2^2. \tag{31}$$

We can get the ranges of $\nu_1^2$ and $\nu_2^2$,

$$
\begin{aligned}
\nu_1^2 &= \left(\frac{\mu_1 + (f-2)\mu_2}{(\mu_1 - \mu_2)(\mu_1 + (f-1)\mu_2)}\right)^2 \le \frac{1}{(\mu_1 - \mu_2)^2} \\
&= \left(\frac{2^f}{(1 + e^{-\frac{1}{\sigma^2}})^{f-2} e^{-\frac{1}{\sigma^2}}}\right)^2 \le 2^{2f} e^{2/\sigma^2}, \\
\nu_2^2 &= \left(-\frac{\mu_2}{(\mu_1 - \mu_2)(\mu_1 + (d-1)\alpha_2)}\right)^2 = \frac{1}{(\mu_1 + (f-1)\mu_2)^2} e^{2/\sigma^2} \\
&= \frac{2^{2f} e^{\frac{2}{\sigma^2}}}{(1 + e^{-\frac{1}{\sigma^2}})^{2f-4}(e^{-\frac{1}{\sigma^2}} + f)^2} \le 2^{2f} f^{-2} e^{2/\sigma^2}.
\end{aligned}
\tag{32}
$$

Thus, the upper bound of $\left\|\boldsymbol{B}^{-1}\right\|_F$ is:

$$
\left\|\boldsymbol{B}^{-1}\right\|_F \le \sqrt{2^{2f} f e^{2/\sigma^2} + (f^2 - f) f^{-2} 2^{2f} e^{2/\sigma^2}} \le (f+1)^{\frac{1}{2}} 2^f e^{1/\sigma^2}.
\tag{33}
$$

The second step presents the specific form of the $\boldsymbol{C}$ matrix,

$$
\boldsymbol{C} =
\begin{bmatrix}
\frac{1}{m}\sum_i \rho(z_{i1})^2 & \frac{1}{m}\sum_i \rho z_{i1} z_{i2} & \cdots & \frac{1}{m}\sum_i \rho z_{i,1} z_{if} \\
\frac{1}{m}\sum_i \rho z_{i1} z_{i2} & \frac{1}{m}\sum_i \rho(z_{i2})^2 & \cdots & \frac{1}{m}\sum_i \rho z_{i2} z_{if} \\
\vdots & \vdots & \ddots & \vdots \\
\frac{1}{m}\sum_i \rho z_{i1} z_{if} & \frac{1}{m}\sum_i \rho z_{i2} z_{if} & \cdots & \frac{1}{m}\sum_i \rho(z_{if})^2
\end{bmatrix},
\tag{34}
$$

each element of the $\boldsymbol{C}$ matrix is less than or equal to $\frac{\rho f}{m}$, then we have $\|\boldsymbol{C}\|_F \in \left[0, f^2 \rho m^{-1}\right]$.

Eventually, for the matrix $\boldsymbol{E}$, which is a $r \times r$ matrix where each element is $\frac{1}{m}$, we have $\|\boldsymbol{E}\|_F = \frac{r}{m}$.

## B.2. Iterative optimal solution for $\boldsymbol{A}'^\top$

By the banach fixed point theorem, there exists self-mapping of $\boldsymbol{A}'^\top$ denoted as $T(\boldsymbol{A}_1'^\top)$ or $T(\boldsymbol{A}_2'^\top)$, which satisfies the conditions of the theorem.

$$
\begin{aligned}
\left\|T\left(\boldsymbol{A}_1'^\top\right) - T\left(\boldsymbol{A}_2'^\top\right)\right\|_F &\le \left\|\boldsymbol{B}^{-1}\right\|_F \|\boldsymbol{C}\|_F \|\boldsymbol{E}\|_F \left\|\boldsymbol{A}_1'^\top - \boldsymbol{A}_2'^\top\right\|_F, \\
\|\boldsymbol{C}\|_F &\in \left[0, \frac{\rho f^2}{m}\right], \|\boldsymbol{E}\|_F = \frac{r}{m}, \\
\pi(\cdot) &\in [0,1], \boldsymbol{Z} \in \{0,1\}^{m \times f}, \\
\left\|\boldsymbol{B}^{-1}\right\|_F &\le (f+1)^{\frac{1}{2}} 2^f e^{1/\sigma^2}.
\end{aligned}
\tag{35}
$$

To ensure that $\left\|\boldsymbol{B}^{-1}\right\|_F \|\boldsymbol{C}\|_F \|\boldsymbol{E}\|_F < 1$, the condition on the condition on $\rho$ is: $\rho < 2^{-f}(f+1)^{-\frac{1}{2}} m^2 f^{-2} r^{-1} e^{-\frac{1}{\sigma^2}}$. This inequality guarantees that the product of the Frobenius norms $\left\|\boldsymbol{B}^{-1}\right\|_F$, $\|\boldsymbol{C}\|_F$, and $\|\boldsymbol{E}\|_F$ is less than 1.

So, if Assumption 3.1 holds and $\boldsymbol{Z}$ follows the $\{0,1\}$ distribution, ignoring the regularization constraints, the condition for convergence is given by: $\rho < 2^{-f}(f+1)^{-\frac{1}{2}} m^2 f^{-2} r^{-1} e^{-\frac{1}{\sigma^2}}$, there exists an iterative fixed point $\boldsymbol{A}'^*$ that minimizes the loss.

Let $\boldsymbol{Z}'^\top \boldsymbol{\Pi}\left(\boldsymbol{Z}'\boldsymbol{A}'^\top - F(h(\boldsymbol{Z}))\right) + \boldsymbol{Z}'^\top \boldsymbol{u}\mathbf{1}_{r\times 1}^\top = 0$, then,

$$
\begin{aligned}
\boldsymbol{A}'^\top &= \left(\boldsymbol{Z}'^\top \boldsymbol{\Pi} \boldsymbol{Z}'\right)^{-1}\left(\boldsymbol{Z}'^\top \boldsymbol{\Pi} F(h(\boldsymbol{Z})) - \boldsymbol{Z}'^\top \boldsymbol{u}_{1\times r}\right), \\
\text{s.t.}\quad &\boldsymbol{Z}'\boldsymbol{A}'^\top \times \mathbf{1}_{r\times 1} = \mathbf{1}_{m\times 1}.
\end{aligned}
\tag{36}
$$

## B.3. Proof of convergence

Matrix Hoeffding's Inequality: Let $\boldsymbol{M_i}$ be an independent, random symmetric matrix of dimension $\Phi$. Let $\boldsymbol{V_i}$ represent a sequence of fixed symmetric matrices, assume that each random matrix satisfies the following conditions:

$$
\mathbb{E}\left[\boldsymbol{M_i}\right] = 0 \quad \text{and} \quad \boldsymbol{M_i}^2 \preccurlyeq \boldsymbol{V_i}^2 \quad \text{almost surely.}
\tag{37}
$$

For any $t \geq 0$,

$$\mathbb{P}\left(\lambda_{\max}\left(\sum_{i=1}^{n} \boldsymbol{M_i}\right) \geq t\right) \leq \Phi \cdot \exp\left(\frac{-t^2}{8\sigma^2}\right), \tag{38}$$

where $\sigma^2 := \left\|\sum_{i=1}^{m} \boldsymbol{V_i^2}\right\|_{\mathrm{op}}$. Let $\boldsymbol{v} \in \mathbb{R}^D$ such that $\|\boldsymbol{v}\| = 1$, we have

$$\left|(\boldsymbol{Mi_v})_j\right| = \sum_{k=1}^{\Phi} (\boldsymbol{M_i})_{j,k}\, v_k \leq \left\|(\boldsymbol{M_i})_j\right\| \cdot \|\boldsymbol{v}\| \leq \sqrt{\Phi}. \tag{39}$$

This leads to $\|\boldsymbol{M_i v}\| \leq \Phi$, and for any $i$, we have $\|\boldsymbol{M_i}\|_{op} \leq \Phi$. Therefore, if we choose $\boldsymbol{V_i} = \Phi\boldsymbol{I_\Phi}$, we obtain

$$\mathbb{P}\left(\lambda_{\max}\left(\sum_{i=1}^{m} \boldsymbol{M_i}\right) \geq t\right) \leq \Phi \cdot \exp\left(\frac{-t^2}{8m\Phi^2}\right), \tag{40}$$

by exchanging $\boldsymbol{V_i}$ and $\boldsymbol{M_i}$, we have $\boldsymbol{M_i} \preccurlyeq \boldsymbol{V_i}$, which implies $\boldsymbol{M_i^2} \preccurlyeq \boldsymbol{V_i^2}$. Assuming that each element of $\boldsymbol{M_i}$ satisfies $(\boldsymbol{M_i})_{j,k} \in [-1, 1]$, we can then conclude that for $\forall t \geq 0$,

$$\mathbb{P}\left(\left\|\frac{1}{m}\sum_{i=1}^{m} \boldsymbol{M_i}\right\|_{\mathrm{op}} \geq t\right) \leq 2\Phi \cdot \exp\left(\frac{mt^2}{8\Phi^2}\right). \tag{41}$$

Then, for each element of $\boldsymbol{\Pi Z Z}^\top$, it belongs to $[0, 1]$, and similarly, for each element of the weight matrix $\pi \in [0, 1]$, the same holds for each element of $\boldsymbol{\Delta}$. We define,

$$\boldsymbol{M_i} := \boldsymbol{\Pi Z Z}^\top - \boldsymbol{\Delta}, \tag{42}$$

in this case, the assumption is satisfied, and since $\Phi = \phi + 1$, we have $\frac{1}{m}\sum_{i=1}^{m} \boldsymbol{M_i} = \widetilde{\boldsymbol{\Delta}} - \boldsymbol{\Delta}$. Therefore, $\forall t \geq 0$,

$$\mathbb{P}\left(\|\widetilde{\boldsymbol{\Delta}} - \boldsymbol{\Delta}\|_{\mathrm{op}} \geq t\right) \leq 4\phi \cdot \exp\left(\frac{-mt^2}{32\phi^2}\right). \tag{43}$$

Since each element of $\boldsymbol{\Delta_m} - \widetilde{\boldsymbol{\Delta}}$ is within the range $\left[-\frac{1}{4}, 1\right]$, for $\forall t \geq 0$, we obtain

$$\mathbb{P}\left(\left\|\boldsymbol{\Delta_m} - \widetilde{\boldsymbol{\Delta}}\right\|_F \geq t\right) \leq 2f \exp\left(-\frac{mt^2}{8f^2}\right), \tag{44}$$

similarly, since $\boldsymbol{\Gamma_m} - \widetilde{\boldsymbol{\Gamma}}$ is also in matrix form, with each element in the range $[-u, 1-u] \subseteq [-1, 1]$, for $\forall t \geq 0$, we have

$$\mathbb{P}\left(\left\|\boldsymbol{\Gamma_m} - \widetilde{\boldsymbol{\Gamma}}\right\|_F \geq t\right) \leq 2f \exp\left(-\frac{mt^2}{8f^2}\right), \tag{45}$$

let $t_1 = 2^{-(1+f)}(f+1)^{-\frac{1}{2}}f^{-\frac{1}{2}}e^{-\frac{1}{\sigma^2}}\epsilon$, $m_1 = 2^{5+2f}(f+1)f^3 e^{\frac{2}{\sigma^2}}\epsilon^{-2}\ln\left(\frac{4f}{\delta}\right)$, we obtain

$$\mathbb{P}\left(\left\|\widetilde{\boldsymbol{\Delta}}^{-1}\right\|_F \left\|\boldsymbol{\Gamma_m} - \widetilde{\boldsymbol{\Gamma}}\right\|_F \geq \frac{\epsilon}{2}\right) \leq \frac{\delta}{2}, \tag{46}$$

similarly, $t_2 = 2^{-1}f^{-\frac{1}{2}}r^{-\frac{1}{2}}\epsilon$, $m_2 = 2^5 f^{\frac{5}{2}}r^{\frac{1}{2}}\epsilon^{-2}\ln\left(\frac{4f}{\delta}\right)$, we obtain

$$\mathbb{P}\left(\left\|\boldsymbol{\Delta_m}^{-1} - \widetilde{\boldsymbol{\Delta}}^{-1}\right\|_F \|\boldsymbol{\Gamma_m}\|_F \geq \frac{\epsilon}{2}\right) \leq \frac{\delta}{2}, \tag{47}$$

therefore, when $m = \max(m_1, m_2)$, we have

$$\mathbb{P}\left(\left\|\boldsymbol{\Delta_m}^{-1}\boldsymbol{\Gamma_m} - \widetilde{\boldsymbol{\Delta}}\widetilde{\boldsymbol{\Gamma}}\right\|_F \leq \epsilon\right) \geq 1 - \delta. \tag{48}$$

It is worth noting that the theorem provides a default statement for the theoretical realization of LIMEFLDL. The key distinction is that regularization terms are not considered. Additionally, $Theorem$ 3.3 holds under fairly mild assumptions about the model $F$ to be explained. Essentially, we only require that $F$ is bounded on the bins. If these bins are computed from a finite training set $\mathcal{X}$, then $Z$ is compact. We essentially require that $F$ is well-defined on $Z$. This condition is almost always satisfied for most machine learning models.

**Limitations.** The strongest limitation of $Theorem$ 3.3 is the poor concentration for small bandwidths. In fact, for small $\sigma$ (i.e., $\sigma < 1$), $Theorem$ 3.3 requires the use of $n$-degree polynomials in $F$ to achieve convergence. This often means that, once the dimensionality exceeds 10, applying $Theorem$ 3.3 in practice becomes very challenging for $\sigma < 1$. In such cases, the effects of regularization start to play a key role. However, as long as $f > 10$ and the default bandwidth, the performance of $Theorem$ 3.3 in LIMEFLDL remains highly satisfactory. On the other hand, as $\sigma \to 0$ it can be directly shown that $A \to 0$ for any $1 \leq j \leq f$. In the extremes, the behavior of the interpretable coefficients is not straightforward. In particular, the sign of the interpretable coefficients can change as the bandwidth varies. This is a concerning phenomenon: for certain values of the bandwidth $\sigma$, the explanations provided by LIMEFLDL can become zero.

## B.4. Proof of stability

For any two similar black-box models $P$ and $Q$, define $T\left(h\left(\boldsymbol{Z}\right)\right) = |P\left(h\left(\boldsymbol{Z}\right)\right) - Q\left(h\left(\boldsymbol{Z}\right)\right)|$, and $T\left(h\left(\boldsymbol{Z}\right)\right)$ of each element $t\left(h\left(\boldsymbol{Z}\right)\right) < \epsilon$. To compare two similar black-box models, we first obtain their explanatory feature attribution matrices, $\boldsymbol{A}'_{\boldsymbol{P}}$, and $\boldsymbol{A}'^{\top}_{\boldsymbol{Q}}$. Then, we calculate the distance between these matrices,

$$\begin{aligned} \left\| \boldsymbol{A}'_{\boldsymbol{P}} - \boldsymbol{A}'^{\top}_{\boldsymbol{Q}} \right\|_F &= \left\| \left(\boldsymbol{Z}'^{\top}\boldsymbol{\Pi}\boldsymbol{Z}'\right)^{-1} \left(\boldsymbol{Z}'^{\top}\boldsymbol{\Pi}T(h(\boldsymbol{Z}))\right) \right\|_F \\ &\leq \|\boldsymbol{\Delta}^{-1}\|_F \|\boldsymbol{Z}'^{\top}\boldsymbol{\Pi}T(h((\boldsymbol{Z}))\|_F, \end{aligned} \tag{49}$$

where $\left\|\boldsymbol{\Delta}^{-1}\right\|_F \leq (f+1)^{\frac{1}{2}} 2^f e^{1/\sigma^2}$, and that each element of $\boldsymbol{Z}'^{\top}\boldsymbol{\Pi}T(h((\boldsymbol{Z}))$ is $m\pi z\epsilon$, where the expected value of $\pi z$ is less than $\frac{1}{2}$, we can conclude that when $\lambda \in \left[0, m2^{f-1}(f+1)^{\frac{1}{2}} f^{\frac{1}{2}} r^{\frac{1}{2}} e^{\frac{1}{\sigma^2}} \epsilon\right]$, the following holds: $\left\|\boldsymbol{A}'_{\boldsymbol{P}} - \boldsymbol{A}'^{\top}_{\boldsymbol{Q}}\right\|_F \leq \lambda$.

## B.5. Proof of property

From a more practical perspective, considering the properties of additive feature attribution, let us assume that our knowledge of $F$ is imperfect. More precisely, we can decompose the function into two parts for explanation, the part originating from the black-box model $F$ itself, and the part arising from perturbations.

Let $\boldsymbol{A}'^{\top}$ represent the obtained interpretation, $\xi_{ki}$ be the feature originally located at the $k$-th position. By replacing it with $\xi_{kj}$ and ensuring that the output of the interpreted model remains consistent, we obtain the following:

$$(\xi_1, \xi_2 \ldots \xi_{ki} \ldots \xi_f) \, \boldsymbol{A}'^{\top} = (\xi_1, \xi_2 \ldots \xi_{kj} \ldots \xi_f) \, \boldsymbol{A}'^{\top}, \tag{50}$$

based on this, we can obtain the following:

$$\xi_{ki} a_{kt} = \xi_{kj} a_{kt}, \forall t \in \{1, 2, \ldots, r\}, \tag{51}$$

where $a_{kt}$ is the explained value of the $t$-th label for the $k$-th feature in the explained feature attribution matrix. As seen above, the $k$-th row of $\boldsymbol{A}'^{\top}$ is all zeros, This means that the feature attribution for all labels of the $k$-th feature are 0. This indicates the process of feature selection. For unselected features, which were not involved in training the local linear model, their feature attributions are all 0. We randomly select two features, $\xi_i$ and $\xi_j$, from the interpretable representation $\xi$, ensuring that $\xi_i \neq \xi_j$. We then exchange these two features to create a new interpretable representation, $\boldsymbol{\xi}'$. Meanwhile, we ensure that the outputs of the interpretation models remain consistent across the different interpretable representations. This results in:

$$(\xi_1, \xi_2, \ldots \xi_i, \ldots \xi_j, \ldots \xi_f) \, \boldsymbol{A}'^{\top} = (\xi_1, \xi_2, \ldots \xi_j, \ldots \xi_i, \ldots \xi_f) \, \boldsymbol{A}'^{\top}. \tag{52}$$

Based on this, we can obtain

$$\xi_i a_{ik} + \xi_j a_{jk} = \xi_i a_{jk} + \xi_j a_{ik}, \forall k \in \{1, 2, \ldots, r\}, \tag{53}$$

based on this random exchange, we can deduce that the $i$-th and $j$-th rows of the feature attribution matrix $A'^{\top}$ are identical.

## B.6. Probabilistic proof of the possibility of negative KL scatter

The KL divergence between two discrete probability distributions $\Psi$ and $\Theta$, defined over the same probability space $\Omega$, is given by:

$$D_{\mathrm{KL}}(\Psi\|\Theta) = \sum_{x\in\Omega} \Psi(x)\log\left(\frac{\Psi(x)}{\Theta(x)}\right), \tag{54}$$

for continuous probability distributions with density functions $\psi(x)$ and $\theta(x)$, the KL divergence is defined as:

$$D_{\mathrm{KL}}(\Psi\|\Theta) = \int_\Omega \psi(x)\log\left(\frac{\psi(x)}{\theta(x)}\right)dx, \tag{55}$$

our goal is to show that $D_{\mathrm{KL}}(\Psi\|\Theta) \geq 0$ is constant when the sum of the probabilities of the predicted distribution is unity. However, $D_{\mathrm{KL}}(\Psi\|\Theta)$ can be less than 0 if the sum of the probabilities of the predicted distribution exceeds 1.

Consider the function $\beta(t) = -\log t$, which is convex for $t > 0$, this follows from its second derivative:

$$\beta''(t) = \frac{1}{t^2} > 0, \tag{56}$$

for a random variable $Y$ and a convex function $\beta$, Jensen's inequality states that:

$$\mathbb{E}[\beta(Y)] \geq \beta(\mathbb{E}[Y]), \tag{57}$$

where $\mathbb{E}$ denote the expected value. We define $Y(x) = \frac{\Theta(x)}{\Psi(x)}$. When $\Psi(x) = 0$, $Y(x)$ is undefined. However, since the points where $\Psi(x) = 0$ do not contribute to the expected value, we can disregard them. Now, we proceed by computing $\mathbb{E}_\Psi[\beta(Y)]$,

$$\begin{aligned}
\mathbb{E}_\Psi[\beta(Y)] &= \sum_{x\in\Omega} \Psi(x)\beta\left(\frac{\Theta(x)}{\Psi(x)}\right) \\
&= \sum_{x\in\Omega} \Psi(x)\left[-\log\left(\frac{\Theta(x)}{\Psi(x)}\right)\right] \\
&= \sum_{x\in\Omega} \Psi(x)\log\left(\frac{\Psi(x)}{\Theta(x)}\right) \\
&= D_{\mathrm{KL}}(\Psi\|\Theta).
\end{aligned} \tag{58}$$

To compute $\beta\left(\mathbb{E}_\Psi[Y]\right)$,

$$\mathbb{E}_\Psi[Y] = \sum_{x\in\Omega} \Psi(x)\left(\frac{\Theta(x)}{\Psi(x)}\right) = \sum_{x\in\Omega} \Theta(x), \tag{59}$$

therefore,

$$\beta\left(\mathbb{E}_\Psi[Y]\right) = \beta(\sum_{x\in\Omega}\Theta(x)) = -\log(\sum_{x\in\Omega}\Theta(x)). \tag{60}$$

According to Jensen's inequality:

$$D_{\mathrm{KL}}(\Psi\|\Theta) = \mathbb{E}_\Psi[\beta(Y)] \geq \beta\left(\mathbb{E}_\Psi[Y]\right) = -\log(\sum_{x\in\Omega}\Theta(x)). \tag{61}$$

When $\sum_{x\in\Omega}\Theta(x) = 1$, we have,

$$-\log(\sum_{x\in\Omega}\Theta(x)) = -\log(1) = 0. \tag{62}$$

However, when $\sum_{x\in\Omega}\Theta(x) > 1$, $-\log(\sum_{x\in\Omega}\Theta(x)) < 0$. Consequently, the KL divergence $D_{\mathrm{KL}}(\Psi\|\Theta)$ is greater than or equal to a number less than 0. Therefore, there is a probability where the KL divergence can be negative. By dividing according to the interval, we find that this probability is given by,

$$\mathbb{P}(\Theta(x)) = \frac{\sum_x \Theta(x) - 1}{\sum_x \Theta(x)}. \tag{63}$$

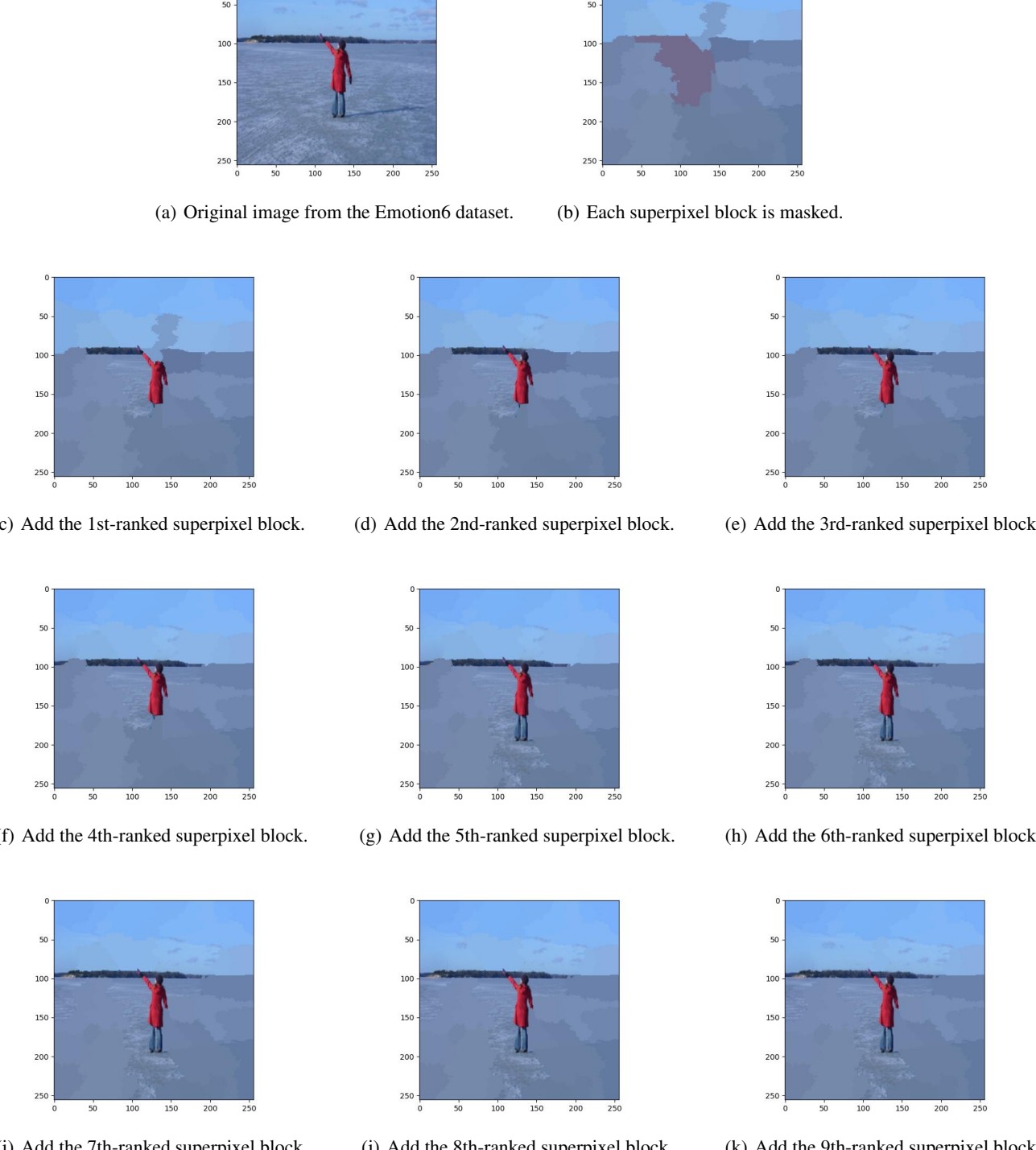

(a) Original image from the Emotion6 dataset.  (b) Each superpixel block is masked.

(c) Add the 1st-ranked superpixel block.  (d) Add the 2nd-ranked superpixel block.  (e) Add the 3rd-ranked superpixel block.

(f) Add the 4th-ranked superpixel block.  (g) Add the 5th-ranked superpixel block.  (h) Add the 6th-ranked superpixel block.

(i) Add the 7th-ranked superpixel block.  (j) Add the 8th-ranked superpixel block.  (k) Add the 9th-ranked superpixel block.

*Figure 11.* Processing of additive attribution of image data.

