# OpenReview forum: "LIMEFLDL: A Local Interpretable Model-Agnostic Explanations Approach for Label Distribution Learning"
_ICML.cc/2025/Conference — ICML 2025 poster_

### Official Review · Reviewer_9QRm · 2025-02-28

**Overall Recommendation:** 4

**Summary:**

Existing interpretability models are designed for single-label paradigm and struggle to directly interpret label distribution learning (LDL) models. To solve this, the paper proposes an improved LIME algorithm capable of effectively interpreting black-box models in LDL. The authors also provide analysis on analytical solution, convergence, stability, and algorithm properties.
## update after rebuttal
I have carefully read the rebuttal. The rebuttal answer the question that how the scoring function in Equation (8) is used to select features and revised some typos. Considering the novelty and the advantage of the proposed method. I keep my socre.

**Claims And Evidence:**

The authors claim that existing interpretability models face challenges in explaining LDL from three aspects: label dependency, computational complexity, and label distribution constraints. These claims are clear and well-supported by evidence.

**Essential References Not Discussed:**

The paper has thoroughly discussed appropriate related work.

**Experimental Designs Or Analyses:**

Yes, the experiments follow a classic comparison protocol, making them sound and valid.

**Methods And Evaluation Criteria:**

Yes, the proposed methods and evaluation criteria make sense for the problem at hand.

**Other Comments Or Suggestions:**

Steps 3, 4 & 8 in Algorithm 1 lack sufficient rigor.

Typos:
1. Line 74: adapt to;
2. Line 117-118: $\mathbb{R}^{r \times 1}$?
3.  Line 159-160: entropy?
4.  Duplicate definition of $\boldsymbol{B}$ in Equations (9) & (10).

**Other Strengths And Weaknesses:**

Strengths:
1. The proposed algorithm is able to interpret any black-box model in LDL.
2. The authors provide an in-depth analysis of the proposed algorithm, including discussions on its analytical solutions, convergence, stability, and overall properties.
3. The paper is clear and has intuitive presentation of the analysis and experiments. For example, the visualization of dummy variable analysis helps in understanding the model’s behavior. Additionally, the introduction of the Jaccard index effectively demonstrates the superiority of the proposed algorithm.

Weakness:
1.	Steps 3, 4 & 8 in Algorithm 1 lack sufficient rigor.
2.	The authors introduce some notations in Equation (9) for convenience, but why are these notations not consistently used in the subsequent paragraphs, such as in Equations (11) & (12)?
3.	There are some typos in this paper, see the next part.

**Questions For Authors:**

1. Could the authors briefly explain how the scoring function in Equation (8) is used to select features, i.e., in Step 8 of Algorithm 1, and whether the selected features are related to the subsequent steps?
2.  The authors introduce some notations in Equation (9) for convenience, but why are these notations not consistently used in the subsequent paragraphs, such as in Equations (11) & (12)?

**Relation To Broader Scientific Literature:**

Currently, there are no other work on interpretable LDL, so I find this paper well-positioned. This work significantly advances the understanding and application of existing LDL models.

**Theoretical Claims:**

I did not check the correctness of any proofs for the theoretical claims, but they appear to be reasonable.

---

> ### Author Rebuttal · Authors · 2025-03-31
>
> Thank you for your positive review of our paper; we greatly appreciate your comments and questions.
> [Comment 1] Steps 3, 4 in Algorithm 1 lack sufficient rigor.
> A: We will introduce Steps 3, 4 of Algorithm 1 in more detail in the revised version. Step 3 of Algorithm 1 generates $m$ samples based on the examples to be interpreted $x$. For image data, the input image is segmented into superpixels, we then generate $m$ samples via binary masking: 0 replaces a superpixel with mean values (per channel), while 1 retains the original pixels, this process preserves local structure while enabling efficient sampling. For tabular data, we first bin features based on $x$’s values (1 if in the same bin as $x$, 0 otherwise), samples are generated by sampling from Gaussian distributions parameterized by training-set statistics (mean/variance per feature). Steps 4 is to calculate pairwise weights between these sampled examples to the example to be interpreted using Eq. 3 and Appendix A.2, ensuring locality-awareness.
>
> [Comment 2] Steps 8 in Algorithm 1 lack sufficient rigor. Could the authors briefly explain how the scoring function in Equation (8) is used to select features, i.e., in Step 8 of Algorithm 1, and whether the selected features are related to the subsequent steps?
> A: In Step 8, we perform iterative feature selection by progressively adding features and evaluating their impact using the scoring function in Eq.8. At each iteration, we train model using only the currently selected features and compare the new score with the current maximum. If the score exceeds the maximum, the feature is retained in the selected set; otherwise, it is discarded. Unselected features are excluded from subsequent training iterations. For validation, we analyze the top 10 ranked features by incrementally adding them and monitoring their influence on explanation fidelity in our local accuracy experiments. In the dummy feature experiment, we search for the current lowest ranked feature and perform masking operation to observe the effect on the interpretation results before and after the change. We will introduce this section in more detail in the revised version to make it easier to understand.
>
> [Comment 3] The authors introduce some notations in Equation (9) for convenience, but why are these notations not consistently used in the subsequent paragraphs, such as in Equations (11) & (12)?
> A: The introduction of symbols in Eq.12, specifically the reformulation of $Z^{\prime \top}{\Pi}F(h(Z))-Z^{\prime \top} u 1_{1 \times r}=\Gamma$, was designed to simplify the algebraic representation of subsequent theorems and proofs. While the original notation $Z^{ \prime \top} \Pi Z^{ \prime}=\Delta$ remains mathematically rigorous, retaining $B$ (instead of $\Delta$) streamlines derivations by reducing nested terms. We acknowledge that this substitution introduces minor notational complexity in the proof flow, and in the revised manuscript, we will add a footnote linking the notations.
>
> [Comment 4] There are some typos in this paper.
> A: In the revised manuscript, we will correct the following issues, Line 74 should be "adapt to" instead of "adopt to", Lines 117-118 should be $a_{i}\in \mathbb{R} ^{r\times 1}$, and Lines 159-160 should be "entropy" instead of "entory". The $B$ matrix of Eq.10 is just a more detailed description of Eq.9, which we will consolidate these equations to explicitly show that matrix $B$.

---

> > ### Comment · Reviewer_9QRm · 2025-04-02
> >
> > Thanks the authors for their efforts in providing the rebuttal. I have carefully read the rebuttal. I still have a minior questions: In Q2, the authors claim that "if the socre exceeds the maximum,...." i wonder how to decide the "maximum"? Can any details can be provided?

---

> > > ### Author Response · Authors · 2025-04-02
> > >
> > > Thank you for this question. This maximum score is iteratively updated, my initial setting is $max=-100000000$, every time a feature is added, the score is calculated using the feature selection scoring function, and then compared to the current $max$, if it is larger than it replaces it with the current score and puts the feature into the set to be selected. If it is smaller than it keep the current $max$.

---

### Official Review · Reviewer_EsHf · 2025-03-04

**Overall Recommendation:** 4

**Summary:**

This paper proposes an interpretability model for label distribution learning (LDL). The classification LIME approach is adapted to handle LDL. An optimization objective is proposed to estimate the parameters of the interpretability model. Theoretical analysis is performed, including convergence, stability, and some theoretical properties. Experimental results show that the proposed method can outperform PULIME, a parallel application of LIME.

**Claims And Evidence:**

The claim that the interpretability model for LDL is important is good. However, it is not clear how the proposed method differs from the classical LIME approach. Although multiple labels are considered for LDL and a single label is considered for LIME, I think they are very similar, since the optimization problem in Eq. (4) can be decomposed for each row of $\boldsymbol{A}\'$, which corresponds to an optimization problem for each label in LIME. In this way, it seems that the proposed method, which mainly works by solving the optimization problem in Eq. (4), is very similar to LIME. The differences are mainly in the different optimization techniques (L-BFGS) and the label distribution constraints (non-zero and normalization).

**Essential References Not Discussed:**

There is no essential reference that needs to be discussed.

**Ethical Review Concerns:**

Since an experiment involves human participants, the ethical statement should describe in detail how the privacy of the participants will be protected and whether the evaluation will be unbiased.

**Ethical Review Flag:**

Flag this paper for an ethics review.

**Ethics Expertise Needed:**

["Responsible Research Practice (e.g., IRB, documentation, research ethics, participant consent)"]

**Experimental Designs Or Analyses:**

- In the first part of the experiments, the fidelity is calculated as several LDL metrics using model outputs from black-box models and the interpretability model. However, the main purpose of the interpretability model is to determine the importance of different features, as described in the previous LIME paper. Therefore, it is uncertain whether a better metric value indicates better interpretability model performance. Therefore, it is recommended that the previous evaluation procedure in the LIME paper can be adopted to validate the effectiveness of the proposed method.

- The second part of the experiments includes human experiments. Human participants are asked to decide whether the explanations are good or bad. It is unclear whether the decisions are objective and influenced by the choice of participants, since only one experiment is conducted.

- Only LIME is chosen as a baseline, which may be inadequate and outdated. More recent interpretability models should be considered for experimental comparisons.

**Methods And Evaluation Criteria:**

The proposed method is valid.

**Other Comments Or Suggestions:**

The writing of the paper can be improved. Here are some minor points:
- The title should read "Explanation" instead of "Explanations".

- There are two periods in line 18-19.

- It is quite confusing with the introduction of Eq. (9) without any descriptions.

- The wording in lines 257-258 is confusing.

- In Property 3.6 and 3.7, the notation of the $k$-th function is inconsistent.

**Other Strengths And Weaknesses:**

Strengths:
- The interpretability of LDL is a good but under explored topic in the literature. Therefore, the research problem is beneficial and interesting.

- The theoretical analysis is good and validate the effectivness of the proposed method.

Weaknesses:
- The novelty of the proposed method seems limited, since the optimization problem is quite similar to a multiple-label version of LIME. Therefore, the novelty and contributions should be clarified in detail.

- It is unclear whether the experiments in the paper can help validate the effectiveness of the proposed method. It can be observed that the interpretability model is more accurate in fitting the black-box model locally. However, it does not mean that feature importance is well described by the proposed method.

- It is unclear how label dependencies are accounted for in the proposed method, as noted in the Introduction section.

**Questions For Authors:**

- The $\pi$ function is different from the previous LIME method. In LIME, the anchor is the selected example $z$. In this paper, however, it is an all-one vector instead. I wonder why the function is changed in the paper.

- I am confused by Figure 3. I do not know the meaning of the purple color that should be specified in the paper.

**Relation To Broader Scientific Literature:**

N/A.

**Theoretical Claims:**

The theoretical claims are good and support the proposed method. However, the structure of the proofs is unclear, and the proof of Theorem 3.2 is missing.

---

> ### Author Rebuttal · Authors · 2025-03-31
>
> Thank you for your constructive review of our paper.
> [Comment 1]About the structure of the proofs.
> A:We will restructure proofs in the revised version. The proof of Theorem 3.2 is appeared in Appendix B.2 (lines 1025~1040).
>
> [Comment 2]The method's novelty appears limited as it resembles a multi-label LIME extension. While targeting LDL (multi-label), Eq.4 decomposes into per-label subproblems akin to LIME's single-label optimization.
> A:Equation 4 can be decomposed into each row for single-label LIME, which means that our method is a generalized version of LIME. However, combining multiple independent LIME cannot directly give rise to our method. The reasons can be summarized as follows.
> 1. Label dependency. Interpretation of LDL requires handling label dependency, whereas the direct combination of multiple LIMEs cannot handle label dependency since per-label training produces disconnected weight vectors, and inability to consider the simultaneous impact of features on multiple labels. Our approach evaluates features’ impacts simultaneously across all labels from the weight matrices, revealing dependencies.
> 2. Theoretical guarantees. We provide the following theoretical guarantees for our method, as the number of sampled examples increases, our interpretation results provably converge under the label distribution. When the black box model turbulence is smaller, the more stable the interpretation results are, however, these guarantees do not hold for applying multiple single-label LIMEs to approximate label distribution.
> 3. Feature selection. Interpretation of LDL requires consideration of global representational features, and multi-LIME fails to achieve due to its single-label focus (label-specific features). Our method uses distribution information to prioritize features globally relevant features.
>
> [Comment 3]LDL metrics may not directly assess interpretability quality.
> A:We conducted supplementary experiments using R²-score aligned with LIME metric and expanded baselines to GLIME and DLIME, all evaluated under default parameters.
> |Model|Datasets|LIME|DLIME|GLIME|LIMEFLDL|
> |-|-|-|-|-|-|
> ||1|.16|.15|.16|.67|
> ||2|.15|.15|.16|.67|
> ||3|.48|.35|.34|.58|
> ||4|.39|.33|.34|.40|
> |LDL-SCL|5|.34|.25|.25|.30|
> ||6|.37|.34|.32|.38|
> ||7|.94|-|.95|.99|
> ||8|.89|-|.89|.98|
> ||1|.17|.17|.16|.99|
> ||2|.16|.15|.17|.99|
> ||3|.34|.33|.33|.36|
> ||4|.25|.31|.29|.36|
> |RBF-LDL-LRR|5|.26|.47|.36|1|
> ||6|.52|.49|.47|.99|
> ||7|.93|-|.96|.99|
> ||8|.67|-|.94 |.99|
>
> These experiments will be included in revision (space permitting).
>
> [Comment 4]About the decisions are objective and influenced by the choice of participants.
> A: While single-study generalizability is limited, our design mitigated bias via: random recruitment (ML/non-ML backgrounds), independent evaluation of 20 randomized interpretation results, and high inter-rater agreement despite participant diversity.
>
> [Comment 5]About how label dependencies are accounted for.
> A:Label dependencies directly influence the equilibrium states of the parameters $u$ and $\rho$. These parameters jointly govern the weight matrix $A$, adjusting its entries to reflect label correlations, this ensures global consistency, features impacting correlated labels exhibit coordinated weight changes in $A$, perturbing a feature’s weight reveals its systemic effect on the entire label distribution (e.g., simultaneous probability shifts for co-dependent labels).
>
> [Comment 6]About some writing errors.
> A:We will change "Explanations" to "Explanation" and correct the problem with the two periods in the revised version.
>
> [Comment 7]About some inadequate or incorrect descriptions.
> A:We will add intuition before Eq.9 to clarify its role in streamlining Eq.10 and simplifying proofs. Clarify perturbation analysis (lines 257-258): black-box model (BM) perturbations (models $P$ vs. $Q$) impact interpretations. $T(h(Z))$ quantifies output distribution divergence between BM (models $P$ vs. $Q$), while $t(h(Z))$ describes the difference in the degree of descriptiveness of single label of the model outputs. Interpretation stability $\propto (\text{BM perturbation magnitude})^{-1} $. We will describe this part in more detail in the revised version. Property 3.6 uses superscripts ($\xi_{k }^i$, $\xi_{k }^j$) for distinct values of feature $k$. Property 3.7 uses ($\xi_{i}$, $\xi_{j}$) to denote different features. We will correct this part in the revised version.
>
> [Comment 8]About the $\pi$ function.
> A:For interpretation, the example to be interpreted $x$ is mapped to an all-ones vector in the interpretable space. Weights are computed via similarity between $x$'s and sampled instances' representations, aligning with prior work.
>
> [Comment 9]About the purple color in Figure 3.
> A:Fig.3 uses a color scheme: Purple (blue+pink blend) indicates LIME$\approx $LIMEFLDL; blue for LIME$>$LIMEFLDL; pink for LIMEFLDL$>$LIME. Detailed analysis will be discussed in the revised version.

---

### Official Review · Reviewer_UXYg · 2025-03-11

**Overall Recommendation:** 4

**Summary:**

In order to mitigate the interpretability challenge inherent in most label distribution learning (LDL) algorithms when applied to risk-sensitive decision-making scenarios, this paper introduces a novel local interpretable model-agnostic explanation framework specifically tailored for LDL. This approach takes into account the label distribution within the local region and constructs local linear models to effectively approximate the global behavior of the black-box LDL model. Furthermore, the paper conducts a thorough theoretical analysis of the proposed methodology, offering a theoretical assurance that the interpretations it yields in the context of LDL tasks are closely aligned with the actual decision-making process.

**Claims And Evidence:**

The claims made in the submission  are supported by clear and convincing evidence.

**Essential References Not Discussed:**

I did not find the essential missing references in this paper.

**Experimental Designs Or Analyses:**

I have checked the experiments. Specifically, I have examined the adequacy of the dataset in Section 4.1, the completeness of the evaluation metrics in Section 4.2, and the rationality of the experimental procedure in Section 4.3.

**Methods And Evaluation Criteria:**

The integration of label distribution constraints into LIME’s framework is novel. The feature attribution matrix and ADMM-based optimization are appropriate.
Evaluation: Fidelity metrics and consistency are well-chosen.

**Other Comments Or Suggestions:**

First, the word “Explanations” in the title “A Local Interpretable Model-Agnostic Explanations Approach” should be amended for accuracy. Second, the quotation marks on line 566 should be used correctly.

**Other Strengths And Weaknesses:**

Strengths：
The problem addressed in this paper holds substantial significance, as label distribution learning algorithms are often complex and their decision-making correctness is challenging to validate in practical applications. This paper endeavors to resolve the interpretability issues inherent in label distribution learning, thereby significantly contributing to the expansion of the applicability of the label distribution learning paradigm. Furthermore, the adaptation of traditional machine learning explanation methods to the label distribution learning framework is non-trivial owing to the high-dimensional and interdependent label space.

Weaknesses：
This paper also has some limitations. For example, the writing of this paper needs improvement, and the experimental results in appendix need more discussions.

**Questions For Authors:**

In Figure 6, what are the differences between the original image and the processed image? Visually, they appear almost same. Besides, are there detailed discussions for the visualized cases to demonstrate the performance of LIMEFLDL?

**Relation To Broader Scientific Literature:**

The related literature is interpretable machine learning. The main contributions of this paper can be treated as an extension of traditional LIME (local interpretable model-agnostic explanations) method. Specifically, it adapts LIME to the $r$-dimensional label space with $r-1$ degrees of freedom.

**Theoretical Claims:**

I have checked the correctness of Theorem 3.2, Theorem 3.3, Theorem 3.4, Property 3.5, Property 3.6, and Property 3.7.

---

> ### Author Rebuttal · Authors · 2025-03-31
>
> Thank you for your positive review of our paper; we greatly appreciate your comments and questions.
>
> [Comment 1] This paper also has some limitations. For example, the writing of this paper needs improvement, and the experimental results in appendix need more discussions.
> A: We will follow up by scrutinizing the paper for writing issues, as well as discussing the experimental results in the appendix in more detail.
>
> [Comment 2] First, the word “Explanations” in the title “A Local Interpretable Model-Agnostic Explanations Approach” should be amended for accuracy. Second, the quotation marks on line 566 should be used correctly.
> A: "Explanations" will be changed to "Explanation", the quotes used in line 566 are incorrect and that these essay writing will be corrected in the revised version.
>
> [Comment 3] In Figure 6, what are the differences between the original image and the processed image? Visually, they appear almost same. Besides, are there detailed discussions for the visualized cases to demonstrate the performance of LIMEFLDL?
> A: The processed image in Fig.6 was changed by smudging and adding a green border change the lowest ranked hyperpixel block, which is not conspicuous enough because of its low rank and is often located in the edge region of the image. For both the image dataset and the table dataset we have done a visual comparison of the LIMEFLDL and the original LDL interpretation results, which we will show in the revised version.

---

> > ### Comment · Reviewer_UXYg · 2025-04-07
> >
> > Thanks for your detailed answers. All my questions have now been answered satisfactorily. I have decided to maintain my original rating.

---

### Official Review · Reviewer_3Lj7 · 2025-03-13

**Overall Recommendation:** 4

**Summary:**

To address the local interpretability issues of label distribution learning (LDL), this paper proposes an improved LIME algorithm, namely LIMEFLDL. The algorithm is mainly manifested in three aspects: first, by introducing a feature attribution matrix to address the label dependency issue in LDL tasks; second, by minimizing the output differences between the black-box model and the explanation model within the generated local region to reduce computational complexity; third, by incorporating linear constraints and penalty functions to ensure that the predictions of the explanation model align with the distribution of the labels. In addition to the above, the article provides extensive theoretical proofs regarding the stability and convergence of the algorithm, as well as its analytical solution form. The effectiveness of the algorithm is demonstrated through multiple experiments and human experiments.

**Claims And Evidence:**

Yes

**Essential References Not Discussed:**

There is no any essential references not discussed.

**Experimental Designs Or Analyses:**

Yes. I check the soundness and validity of all experimental designs and analyses.

**Methods And Evaluation Criteria:**

Yes

**Other Comments Or Suggestions:**

1.It is recommended to include statistical information in the comparison results between LIMEFLDL and PULIME, such as how many datasets showed better performance of LIMEFLDL compared to PULIME under the KL metric.

2.In Table 1, what does the abbreviation RBF-LDL-LRR stand for? It is not found in the original text. Additionally, the text in Figure 10 is not clear.

3.In the abstract, on the 22nd line, "To address the label dependency problem," the introduction of the label dependency problem is not sufficiently clear and can be confusing. Adding some rationale would improve this.
Regarding the initial value for the feature attribution matrix A, these are not found in the paper. This information could be added for completeness.

**Other Strengths And Weaknesses:**

Strengths:

1.The authors provide proofs regarding the stability and convergence of the explanation algorithm, thereby increasing the credibility and correctness of the algorithm.

2.The authors compare the LIMEFLDL and PULIME algorithms on 8 datasets across 7 metrics, fully demonstrating the effectiveness of the proposed method.

Weakness:

1.There are some errors and unclear elements regarding mathematical symbols in the paper, which are detailed in the Theoretical Claims section.

2.The captions for figures and tables in the paper are not sufficiently detailed. For example, in Figure 1, the legends are not explained in caption.

**Questions For Authors:**

1.The Introduction mentions the computational complexity issues of the LIMEFLDL algorithm compared to parallel use of the traditional LIME algorithm (PULIME). Is there any theoretical or experimental proof to support this?

2.Regarding the Y_mean in Equation 8, could it be influenced by the class distribution? It might be worth trying to adopt a class-related prior probability distribution.

**Relation To Broader Scientific Literature:**

This paper proposes an improved LIME[1] algorithm, namely LIMEFLDL, to address the local interpretability issues of label distribution learning. The LIME algorithm is designed for single-label learning tasks. However, due to the label dependency in label distribution learning (LDL), LIME is unsuitable for direct application in LDL tasks. This paper introduces the feature attribution distribution matrix to address this issue.

[1] Ribeiro M T, Singh S, Guestrin C. " Why should i trust you?" Explaining the predictions of any classifier[C]//Proceedings of the 22nd ACM SIGKDD international conference on knowledge discovery and data mining. 2016: 1135-1144.

**Theoretical Claims:**

1.For Equation 23 and 24, in the third equation, where does the 2 to the power of f come from?

2.For Equation 64, does this expression satisfy the probability constraint, i.e., the sum of probabilities equals 1? Please provide a more detailed explanation.

3.For Equation 49, the meaning of the sigma symbol has not been explained. Please check if it is written correctly.

4.For Equation 39, there is a symbol error: the symbol u seems to be incorrect.

---

> ### Author Rebuttal · Authors · 2025-03-31
>
> Thank you for your detailed review of our paper; we greatly appreciate your comments and questions.
> [Comment 1]About the errors and unclear mathematical symbols.
> A:
> 1. Equation 23 should be modified: $\sum_{k=0}^{f} e^{\frac{(k-f)}{\sigma^{2}}}\frac{k}{f} \frac{f!}{2^{f}(f-k)!k!}$, this equation is to represent the probability of picking $k$ non-zero elements from $f$ elements, $\frac{f!}{(f-k)!k!}$ is the number of combinations to select $k$ from $f$ elements, $2^{f}$ is the total number of combinations. The same error is found in Eq.24. We will correct them in the revised version.
> 2. What we want to describe is that if $\sum_{x \in \Omega} \Theta(x)>1$, Jensen's inequality (Eq.61) implies $E(KL)\ge-\log (\sum_{x \in \Omega} \Theta(x))<0$. Thus, if $KL\in (-\log (\sum_{x \in \Omega} \Theta(x)),0)$ when distributions violate normalization, the bound's numerator ($\sum_{x}\Theta(x)-1$) and denominator ($\sum_{x } \Theta(x)$) directly quantify this violation.
> 3. In Eq.49, we replaced the original symbol $\Sigma$ with $\Delta$ to avoid ambiguity with the concatenation operator $\Sigma$ used earlier. We'll correct it in the revised version.
> 4. It should be a $v$-vector instead of $u$, we will correct it in the revised version.
>
> [Comment 2]The captions for figures and tables in the paper are not sufficiently detailed.
> A: We will add more details to describe the figures and tables in the revised version. Fig1a: x-axis: exponentially increasing samples (log scale); Fig1b: x-axis: decreasing divergence between two black-box models, both graphs of y-axis is the Top 20 Jaccard Index. Fig1a demonstrates convergence as Jaccard Index $\uparrow$ with exponentially increasing samples (log-scale x-axis), thus illustrating the convergence of the interpretation algorithm. Fig1b shows Jaccard Index $\uparrow$ when predictive distribution divergence between black-box models $\downarrow$ (x-axis), thus illustrating the stability of the interpretation algorithm.
>
> [Comment 3]About the statistical information in the comparison results between LIMEFLDL and PULIME.
> A: We will give ranking statistics in the revised version, as follows,
> |Model|Algorithm|Cheb.|Clark|Canb.|KL|Cos.|Inter.|Jac.|
> |-|-|-|-|-|-|-|-|-|
> |RBF-LDL-LRR|LIME|1.75|1.5|1.625|2|1.625|1.625|1.875|
> ||LIMEFLDL|1.25|1.5|1.375|1|1.375|1.375|1.125|
> |LDL-SCL|LIME|1.625|1.5|1.5|1.875|1.5|1.5|1.625|
> ||LIMEFLDL|1.375|1.5|1.5|1.125|1.5|1.5|1.375|
> |AA-KNN|LIME|1.625|1.625|1.625|1.5|1.625|1.625|1.75|
> ||LIMEFLDL|1.375|1.375|1.375|1.5|1.375|1.375|1.25|
> |MEM|LIME|1.75|1.75|1.75|1.875|1.75|1.75|1.75|
> ||LIMEFLDL|1.25|1.25|1.25|1.125|1.25|1.25|1.25|
>
> LIMEFLDL leads the rankings for the vast majority of measures, we will add this section in the revised version as space permits.
>
> [Comment 4]About the abbreviation RBF-LDL-LRR, and the text in Figure 10.
> A:We enhanced the LDL-LRR model's feature extraction via Gaussian kernel (renamed RBF-LDL-LRR). Figure 10 tests robustness by masking weakest features, measuring fidelity before/after masking. We will add the name of the modified model and the description to the revised version to make it more accessible.
>
> [Comment 5]About the introduction of the label dependency problem and the initial value for the feature attribution matrix A.
> A:We will clarify label dependence fundamentals in the revised version: LDL handles multi-label samples where labels exhibit co-occurrence patterns (simultaneous changes) and dependency propagation (one label's presence affects others' distributions). Matrix $A $ uses uniform initialization as initialization.
>
> [Comment 6]About the computational complexity of the LIMEFLDL algorithm compared to PULIME.
> A: We analyze the computational complexity of LIMEFLDL and PULIME. For LIMEFLDL, sampling and forming weight matrix $\Pi$ requires $O(m)$. Matrix multiplication contributes $O(mfr)$, and black-box model inference accounts for $O(mrk)$. Element-wise matrix subtraction and F-paradigm operations each add $O(mr)$, with regularization terms contributing $O(fr)$. The total complexity is $O(mfr+mrk+mr+mr+fr+m)$. PULIME's per-label workflow involves sampling ($O(m)$), vector operations ($O(m)$ for subtraction/squaring and $O(mf)$ for multiplication), black-box inference ($O(mrk)$), and regularization ($O(f)$). With parallel computation, its total complexity is $O(mrf+mkr^2+mr+mr+mr+rf)$. PULIME's ridge regression solver has $O(f^3+mf^2)$ complexity, LIMEFLDL uses L-BFGS optimization with $O(T(zr+mf))$ cost, where $z≈5–20$ (stored gradient pairs) and $T$ denotes iterations.
>
> [Comment 7]About the Y_mean in Equation 8, and a class-related prior probability distribution.
> A: The idea of using a uniform distribution is to allow the current feature selection to go beyond the effect of a uniform distribution, but of course it is possible to make different prior distributions for specific datasets so that the selected features are more representative.

---

> > ### Comment · Reviewer_3Lj7 · 2025-04-06
> >
> > Thanks for the rebuttal to address my concern. I have increased my score.

---

### Decision · Program_Chairs · 2025-05-01

**Decision:**

Accept (poster)

**Comment:**

This submission worked on interpretability of label distribution learning and proposed an extension of Local Interpretable Model-agnostic Explanations that can effectively interpret any black-box model in label distribution learning. All the four reviewers consistently agreed it is a good paper and should be accepted. There were some concerns about the human experiments, because human experiment results were reported but essential details were missing (including participant recruitment process, participant population description, experimental protocol details, and participant protections). I have checked the ethics review and rebuttal by myself, and I think the authors have addressed the issues. Therefore, we should accept this submission to be published at ICML 2025.

PS: The authors should give the full name of LIME the first time it appears in the abstract, since the abstract is non-technical and targets at a very broad readership.